# Structural basis of the activation of c-MET receptor

Emiko Uchikawa [1,6], Zhiming Chen [2,4,6], Guan-Yu Xiao[2,5], Xuewu Zhang [1,3✉] & Xiao-chen Bai [1,2✉]

The c-MET receptor is a receptor tyrosine kinase (RTK) that plays essential roles in normal cell development and motility. Aberrant activation of c-MET can lead to both tumors growth and metastatic progression of cancer cells. C-MET can be activated by either hepatocyte growth factor (HGF), or its natural isoform NK1. Here, we report the cryo-EM structures of c-MET/HGF and c-MET/NK1 complexes in the active state. The c-MET/HGF complex structure reveals that, by utilizing two distinct interfaces, one HGF molecule is sufficient to induce a specific dimerization mode of c-MET for receptor activation. The binding of heparin as well as a second HGF to the 2:1 c-MET:HGF complex further stabilize this active conformation. Distinct to HGF, NK1 forms a stable dimer, and bridges two c-METs in a symmetrical manner for activation. Collectively, our studies provide structural insights into the activation mechanisms of c-MET, and reveal how two isoforms of the same ligand use dramatically different mechanisms to activate the receptor.

[1] Department of Biophysics, University of Texas Southwestern Medical Center, Dallas, TX, USA. [2] Department of Cell Biology, University of Texas Southwestern Medical Center, Dallas, TX, USA. [3] Department of Pharmacology, University of Texas Southwestern Medical Center, Dallas, TX, USA. [4]Present address: Hengyang Medical College, University of South China, Hengyang, Hunan, China. [5]Present address: Department of Thoracic Head Neck Medical Oncology, University of Texas MD Anderson Cancer Center, Houston, TX, USA. [6]These authors contributed equally: Emiko Uchikawa, Zhiming Chen. ✉email: Xuewu.Zhang@UTSouthwestern.edu; Xiaochen.Bai@utsouthwestern.edu

C-MET receptor, which belongs to the receptor tyrosine kinase (RTK) family, plays essential roles in controlling a number of critical cellular processes such as cell proliferation, survival, motility, and morphogenesis[1,2]. Aberrant activation of c-MET can lead to both tumor growth and metastatic progression of cancer cells, making it an important drug target for cancer treatments[3,4]. C-MET can be activated by its cognate ligand—hepatocyte growth factor (HGF)[5,6]. It has been proposed that the binding of HGF to c-MET induces the dimerization of c-MET that enables its intracellular kinase domains (KDs) to undergo autophosphorylation[7]. The phosphorylated KD recruits cytosolic effector proteins, leading to activation of downstream signaling pathways.

Structurally, the c-MET receptor is composed of a Semaphorin (SEMA) domain, a plexin-semaphorin-integrin (PSI) domain,

and four consecutive immunoglobulin-plexin-transcription factors (IPT1-4) domains in the extracellular region, a single transmembrane helix (TM), and an intracellular KD (Fig. 1a). Notably, c-MET is initially expressed as a 150 kDa single-chain precursor, which then turns into the mature form through the proteolytic cleavage between Arg307 and Ser308 by the furin protease[8]. The mature form contains the α- (32 kDa) and β-subunits (120 kDa) that are linked by at least three disulfide bonds (Fig. 1a). HGF, the ligand of c-MET, is secreted as a single chain 83 kDa precursor protein containing an N-terminal (N) domain, four consecutive kringle (K1–K4) domains, and a serine protease homology (SPH) domain (Fig. 1a). Similar to c-MET, proteolytic cleavage between Arg494 and Val495 of HGF generates the α and β subunits (57 and 26 kDa, respectively), which are linked by a disulfide bond between Cys487 of α-subunit and

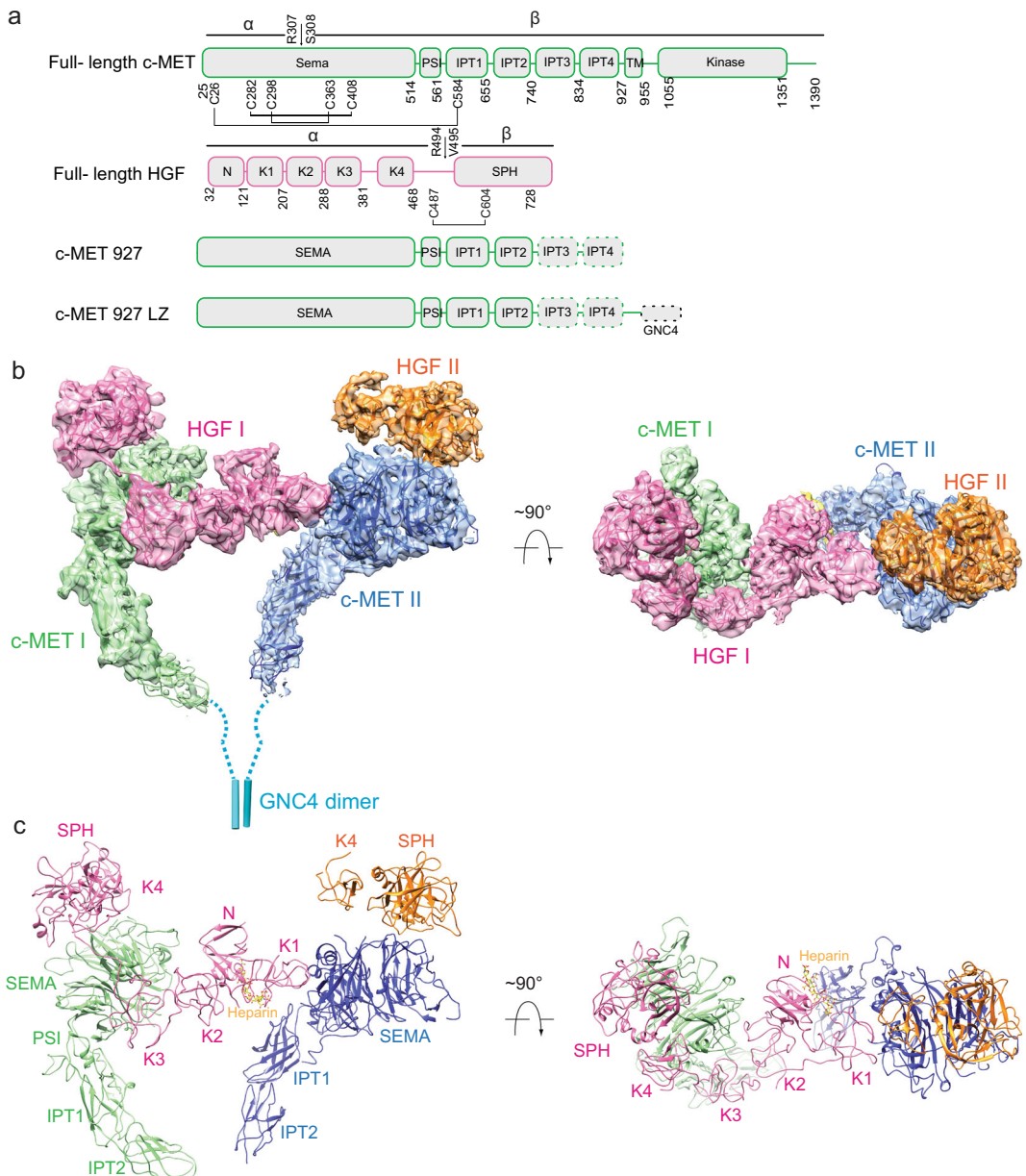

**Fig. 1 Overall structure of the c-MET/HGF holo-complex. a** Domain structures of human c-MET and HGF. The proteolytic cleavage site of c-MET is located between Arg307 and Ser308. The proteolytic cleavage site of HGF is located between Arg494 and Val495. C-MET927-LZ and full-length HGF were used for structural determination in this study. The dash boxes indicate the domains that were unsolved in cryo-EM maps. **b** 3D reconstruction of the 2:2 c-MET/HGF holo-complex and the corresponding ribbon representation of this complex fitted into the cryo-EM map at 4.8 Å resolution, shown in two orthogonal views. **c** The ribbon representation of c-MET/HGF holo-complex shown in two orthogonal views.

Cys604 of β-subunit (Fig. 1a). Interestingly, both of pro-HGF and cleaved HGF can bind to c-MET with high affinity, but only cleaved HGF can activate c-MET signaling, indicating that proteolytic cleavage is critical for HGF-induced c-MET activation[9,10]. It has been further suggested that the proteolytic cleavage of HGF could trigger its structural rearrangement that is likely to be critical for inducing c-MET activation[11,12].

Previous biochemical and structural studies have established that there are at least two distinct c-MET binding sites on HGF[13]. One is mapped to the α-subunit of HGF[10], and the other is located in the SPH domain, the only domain existing in the β-subunit of HGF. The interaction between the SPH domain of HGF and c-MET has been well characterized by X-ray crystallography, showing HGF-SPH makes close contact with the bottom surface of c-MET-SEMA[14]. Although the SPH domain alone cannot activate c-MET, mutations that disrupting the interaction between HGF-SPH and c-MET-SEMA significantly reduce HGF-induced c-MET activation, validating the functional importance of this interaction[15]. Nevertheless, the detailed interaction pattern between the α-subunit of HGF and c-MET remains unknown; thus, the complete picture of how full-length HGF recruits and dimerizes c-MET has not been depicted, hindering our understanding of the exact activation mechanism of c-MET. Furthermore, it has been shown that glycosaminoglycan, such as heparin, heparin sulfate, and dermatan sulfate, can promote HGF-induced c-MET activation through enhancing the binding affinity between HGF and c-MET[16–18]. The heparin-binding site at HGF has been mapped to the N domain by mutagenesis and NMR studies[19,20]. However, the exact heparin-binding site on c-MET has not been identified; therefore, how heparin facilitates the binding of HGF to c-MET is unclear. Finally, c-MET could also be activated by NK1, a naturally occurring splicing variant of HGF that contains only the N and K1 domains of HGF[21]. Different from HGF that exists mainly in a monomeric form, NK1 itself forms a stable dimer that can recruit two c-MET for activation. However, structural details of how NK1 engages and activates c-MET have not been revealed.

Here, we determine the cryo-EM structures of c-MET-HGF and c-MET-NK1 complexes mimicking their active states. The c-MET-HGF complex structure reveals an asymmetric 2:2 c-MET: HGF assembly, in which one of the HGF molecules is able to bridge two c-MET molecules together for receptor activation. This minimal 2:1 c-MET:HGF active complex is further stabilized by the second HGF molecule and heparin, leading to a more stable 2:2 complex and enhanced activation of c-MET. Finally, our structural model of the 2:2 c-MET:NK1 complex shows that distinct from the way that HGF activates c-MET, the NK1 dimer recruits two c-METs in a symmetric manner. The two c-MET molecules bound with the NK1 dimer are placed in close proximity that is ideal for receptor activation. Taken together, our results reveal the activation mechanism of c-MET receptor in response to different types of ligands.

## Results

**Structure determination of c-MET/HGF extracellular complexes**. To understand the activation mechanisms of c-MET by HGF, we successfully expressed and purified both the c-MET extracellular domain (resides: 1–927; denoted as c-MET927) and full-length HGF, and reconstituted the c-MET/HGF complex in vitro, showing excellent behavior in size exclusion chromatography (SEC). However, the resulting cryo-EM map from this sample only showed low-resolution features, indicating the structural flexibility of this complex. This low-resolution structure however showed that the membrane-proximal IPT2 domains of the two c-MET molecules are arranged in close proximity,

consistent with the common ligand-induced dimerization mechanism for RTKs[22]. We reasoned that introducing a leucine zipper motif to the C-terminus of c-MET927 with a short flexible loop (denoted as c-MET927-LZ) could potentially stabilize the c-MET/HGF complex in the active state (Fig. 1a)[23]. As expected, c-MET927-LZ forms a dimer on its own and can bind to HGF in a similar manner as c-MET927 (Supplementary Fig. 1). Tinzaparin sodium, a low molecular weight heparin surrogate, was also added to enhance the binding affinity between c-MET and HGF. Initial data collection using the c-MET927-LZ/HGF sample yielded a cryo-EM map with a similar shape but significantly improved quality as compared to c-MET927/HGF. We, therefore, collected a large dataset for c-MET927-LZ/HGF sample, in order to obtain a high-resolution structure of the c-MET/HGF complex mimicking the active state (Supplementary Figs. 2 and 3).

After the initial 3D classification of c-MET927-LZ/HGF particles set, two major classes were identified (Supplementary Fig. 3). The reconstruction of particles from the first class reached 4.8 Å resolution and showed an asymmetric holo-complex (Supplementary Fig. 2), comprising two c-MET and two HGF molecules (denoted as c-MET I, c-MET II, HGF I, and HGF II). The SEMA and PSI domains of both of c-MET I and II were well resolved in this cryo-EM map, whereas the cryo-EM densities for IPT1 and IPT2 domains were less well defined (Supplementary Fig. 2). The densities of IPT3 and IPT4 domains of two c-MET molecules were completely unresolved in this cryo-EM map, presumably due to structural flexibility. All six domains of HGF I (N, K1–K4, and SPH) showed well-defined densities in the cryo-EM map of holo-complex; however, only K4 and SPH domains of HGF II were resolved, suggesting that the first four domains of HGF II are less tightly bound at the receptor and thus become flexible (Supplementary Fig. 2).

The entire c-MET I as well as the K4 and SPH domains of HGF I were absent in the reconstruction from the second class (sub-complex), which was resolved at 4 Å resolution (Supplementary Figs. 2 and 3). Nevertheless, c-MET II and its bound HGFs, which include N, K1, and K2 domains of HGF I and K4 and SPH domains of HGF II, adopt nearly identical conformation as shown in holo-complex (Supplementary Figs. 2 and 3). We, therefore, modeled the majority part of the c-MET extracellular domain (all domains except for IPT3 and IPT4) as well as the majority part of HGF (all domains except for K3 and K4) by using the cryo-EM structure of sub-complex that was determined at higher resolution, with the help of the available crystal structures of different domains of c-MET and HGF[14,24,25] (Supplementary Fig. 2 and Table 1). Subsequently, we applied a focused refinement approach on the particles used in the reconstruction of holo-complex to improve the resolution for c-MET I and HGF I, resulting in a new cryo-EM map with resolution improved to 4.5 Å, which allowed us to build reliable models for the K3 and K4 domains of HGF (Supplementary Figs. 2, 3 and Table 1). Finally, we could build a model for the majority part of the 2:2 c-MET-LZ/HGF complex by rigid-body fitting all the domains into the cryo-EM structure of holo-complex (Fig. 1b, c).

**Structural model of c-MET/HGF complex in the active state**. The overall structure of the 2:2 c-MET/HGF holo-complex has an asymmetric "Π" shape (Fig. 1b, c). The top part of the Π is composed of the SEMA domains of two c-METs as well as two HGFs; while the lower part of the Π exclusively consists of the IPT1 and IPT2 domains of two c-METs (Fig. 1c). The distance between the two IPT2 domains of c-METs in this structure is ~20 Å, hinting that the two membrane-proximal IPT4 domains of c-METs are likely to be positioned in proximity in this specific

**Table 1 Cryo-EM data collection, refinement, and validation statistics.**

| | c-MET I/HGF I/c-MET II/HGF II EMD-23919 7MO7 | c-MET I/HGF I EMD-23920 7MO8 | c-MET II/HGF I/HGF II (K4, SPH) EMD-23921 7MO9 | c-MET II/HGF I/HGF II (intact) EMD-23922 7MOA | c- MET/NK1 EMD-23923 7MOB |
|---|---|---|---|---|---|
| Data collection and processing | | | | | |
| Magnification | 46,296 | 46,296 | 46,296 | 46,296 | 46,296 |
| Voltage (kV) | 300 | 300 | 300 | 300 | 300 |
| Electron exposure (e$^-$/Å$^2$) | 60 | 60 | 60 | 60 | 60 |
| Defocus range (µm) | 1.08 | 1.6–2.6 | 1.6–2.6 | 1.6–2.6 | 1.6–2.6 |
| Pixel size (Å) | 1.08 | 1.08 | 1.08 | 1.08 | 1.08 |
| Symmetry imposed | C1 | C1 | C1 | C1 | C2 |
| Initial particle images (no.) | 3,287,093 | 3,287,093 | 3,287,093 | 3,287,093 | 1,342,144 |
| Final particle images (no.) | 25,787 | 25,787 | 45,471 | 13,274 | 11,570 |
| Map resolution (Å) | 4.8 | 4.5 | 4 | 4.9 | 5 |
| FSC threshold | 0.143 | 0.143 | 0.143 | 0.143 | 0.143 |
| Refinement | | | | | |
| Initial model used (PDB code) | 2UZX, 1SHY, 1NK1 | 2UZX, 1SHY, 1NK1 | 2UZX, 1SHY, 1NK1 | 2UZX, 1SHY, 1NK1 | 2UZX, 1NK1 |
| Model composition | | | | | |
| Nonhydrogen atoms | 18,138 | 9878 | 9757 | 12,513 | 10,476 |
| Protein residues | 2329 | 1250 | 1247 | 1580 | 1322 |
| Ligands | 1 heparin | 1 heparin | 1 heparin | 2 heparin | |
| R.m.s. deviations | | | | | |
| Bond lengths (Å) | 0.005 | 0.008 | 0.01 | 0.009 | 0.005 |
| Bond angles (°) | 0.992 | 1.299 | 1.486 | 1.626 | 0.767 |
| Validation | | | | | |
| MolProbity score | 2.31 | 2.51 | 2.54 | 2.60 | 2.26 |
| Clashscore | 24.5 | 30.56 | 34.11 | 40.18 | 22.07 |
| Poor rotamers (%) | 0 | 0.09 | 0.38 | 0 | 0 |
| Ramachandran plot | | | | | |
| Favored (%) | 93.32 | 90.23 | 90.79 | 91.25 | 93.49 |
| Allowed (%) | 6.68 | 9.77 | 9.21 | 8.56 | 6.51 |
| Disallowed (%) | 0 | 0 | 0 | 0.19 | 0 |

dimeric conformation that allows the intracellular KDs to undergo *trans*-autophosphorylation. It is worth noting that the C-terminal leucine zipper motif is linked to c-MET with sufficient flexibility that is unlikely to impose the distance restraints between the two c-METs. These structural features strongly suggest that this HGF mediated c-MET dimer represents the active state of c-MET.

Strikingly, in the active structure of the c-MET/HGF dimeric complex, only HGF I simultaneously recruits two c-METs by using two distinct surfaces localized on the opposite sides of this HGF (Fig. 1c). HGF II, however, only engages c-MET II through the interaction between HGF-SPH and c-MET-SEMA; thus, the second HGF is not directly involved in the c-MET activation (Fig. 1c). Such a 2:1 receptor:ligand activation model has been observed in the activation of the growth hormone receptor that belongs to the class I cytokine receptor family[26].

**Structure of the extracellular domain of c-MET**. In the structure of 2:2 c-MET/HGF holo-complex, both c-MET I and II assume nearly identical "9"-shaped conformation (Supplementary Fig. 4a, b). The SEMA domain of c-MET is localized in the head region of "9", while the leg of "9" consists of PSI and the first two IPT domains (Supplementary Fig. 4a). Similar "9"-shaped conformation of c-MET has been observed in the crystal structure of c-MET in complex with bacterial invasion protein InlB[24]. Notably, a small extension segment at the N-terminus of c-MET is completely disordered in the crystal structure of c-MET/InlB. By contrast, the N-terminal segment of c-MET in our structure adopts an extended loop-α conformation and forms extensive interactions with both PSI and IPT1 domains of c-MET (Supplementary Fig. 4a). Moreover, Cys26 from the N-terminal

segment and Cys584 from the IPT1 domain are closely adjacent to each other in our model, suggesting that they form a disulfide bond to further strengthen the interaction between these two structural motifs (Supplementary Fig. 4a). In the previous crystal structure of a c-MET/Fab complex, a short peptide, which is part of the N-terminal segment of c-MET, is bound at the IPT1 domain mainly through the formation of Cys26-Cys584 disulfide bond, consistent with our structural model[27]. A superimposition of our structure with the crystal structure (PDB ID: 2UZX) suggests that, upon the tethering of the N-terminal segment to the PSI and IPT1 domain of c-MET, the entire stalk region of c-MET rotates outward by ~20° (the pivot point is located at the linker between PSI and IPT1 domains), resulting in a more extended conformation (Supplementary Fig. 4c). It is tempting to speculate that the tethering of the N-terminal segment to the stalk region may stabilize c-MET in a specific extended conformation, which might be important for the efficient activation of c-MET. Indeed, if we replace the models of two c-METs from our structure of c-MET/HGF holo-complex to the N-terminus disordered c-MET, the distance between the two IPT2 domains is increased from 20 to 44 Å, suggesting that the compact c-MET is not in an ideal conformation for activation, which supports our hypothesis.

**Structure of c-MET bound full-length HGF**. C-MET-bound HGF I adopts an extended conformation (Supplementary Fig. 4d). At one end of HGF I, the N, K1, and K2 domains (NK2) assemble as a triangular shape, which is virtually identical to that observed in the crystal structure of NK2 alone[25], indicating that NK2 does not undergo any conformational changes upon binding to the c-MET. Nevertheless, no direct interactions were observed between NK2 and K3 domains or between K3 and K4 domains

(Supplementary Fig. 4d). It is, therefore, reasonable to imagine that the relative orientations among NK2, K3, and K4 become fixed only when HGF is bound to c-MET. At the other end of HGF, the SPH domain packs tightly against the K4 domain through extensive domain–domain interaction (Supplementary Fig. 4d). As the K4 and SPH domains belong to the α and β subunits of HGF, respectively, such strong K4/SPH interaction would be important for the structural integrity of HGF α/β heterodimer in addition to the disulfide bond between these two subunits.

**Interactions between c-MET I and HGF I**. As mentioned above, our model shows that one single HGF is sufficient to bridge two c-MET molecules for receptor activation. To achieve this special stoichiometry for c-MET activation, the activating HGF (i.e., HGF I) with an elongated shape simultaneously recruits two c-METs to its opposite ends. In one-half of the holo-complex, N, K2, K3, K4, and SPH domains of HGF I are arranged as a C-clamp shape that packs tightly against the SEMA domain of c-MET I (Fig. 2a). Totally four interfaces are formed between c-MET I and HGF I (denoted as interfaces I, II, III, and IV) (Fig. 2b).

Interface I is formed between a short α-helix in the linker connecting the fifth and sixth blades of c-MET I-SEMA and two short inter-strand loops in the N domain of HGF I (Fig. 2b, c). Specifically, Lys47, Lys91, Phe112, and His114 of HGF interact with Asn393 and Phe398 of c-MET, through van der Waals and hydrogen bonding interactions (Fig. 2c). Consistent with this structural observation, a previous study showed that HGF H114A mutant exhibited ~84% decreased biological activity as compared with wild-type HGF[28]. To further confirm the functional significance of interface I, we mutated the residues Lys47, Lys91, Phe112, and His114 in the N domain of HGF to either glutamate or alanine. HGF WT and mutants were then purified and applied to H1299 cells for 10 min at 37 °C to test their abilities in inducing c-MET activation. As expected, K47E, K91E, F112A, and H114E mutants showed various degrees of reduction in activation of c-MET (Fig. 2f and Supplementary Fig. 5).

Interface II is made between the fourth blade of c-MET SEMA and the K2 domain of HGF I (Fig. 2b, d). A loop of HGF I-K2 (residues 240–253) engages the top surface of the largely twisted β-sheet in the fourth blade of c-MET (Fig. 2d). Three positively charged residues in the loop of HGF I-K2, Arg242, Lys244, and Arg249, interact with negatively charged residues of c-MET, including Glu267 and Asp352; while residues 249–252 of HGF I-K2 make close contact with a surface formed by strands E and F of the fourth blade of c-MET I-SEMA (Fig. 2d). The relevance of the SEMA-K2 interface is supported by previous mutagenesis data showing that mutation of H241, Arg242, Lys244, and Arg249 in HGF significantly reduces its biological activity[19]. We also introduced R242E, K244E, and R249E point mutation or R242E/K244E/R249E triple mutation to the K2 domain of HGF, and showed that all the HGF-K2 mutants exhibited lower potency in activating c-MET as compared with wild-type HGF (Fig. 2f and Supplementary Fig. 5). In addition, we introduced a triple mutation to c-MET-SEMA (E267A/R384A/E419A) to disrupt interface II. This c-MET mutant showed deficiency in HGF-dependent activation (Fig. 2g and Supplementary Fig. 6). Furthermore, our in vitro pull-down binding assay showed that the HGF R242E/K244E/R249E mutant indeed exhibited weaker c-MET binding affinity, as compared to wild-type HGF (Fig. 2h, i). These results further validate the functional significance of interface II for c-MET activation.

Interface III is between the SEMA domain of c-MET I and the K3 domain of HGF I (Fig. 2b, e). This interaction mainly involves a large surface formed together by the fourth, fifth, and sixth blades of c-MET I β-propeller and a flat surface of the K3 domain of HGF I (Fig. 2e). In particular, Trp321 of HGF packs against a short α-helix between strand A and B of the fifth blade of c-MET I-SEMA as well as a short loop that connects strand C and D of the sixth blade of c-MET-SEMA. Glu361 and Tyr376 of HGF I interacts with residues in strand E of the fourth blade of c-MET I-SEMA, including Arg426 and Val427. In addition, Gln309 of HGF makes contact with an α-helix in c-MET I-SEMA, which is localized above the lateral surface of the fourth blade (Fig. 2e). Furthermore, a long loop (residues 300–311) that links the fourth and fifth blades of c-MET I-SEMA adopts an extended conformation, is tethered to the K3 domain of HGF I, which largely contributes to the binding between c-MET I-SEMA and HGF I-K3 (Fig. 2e). This long loop is disordered in the crystal structure of c-MET/InlB complex[24], but displays well-defined density in our cryo-EM map of c-MET/HGF holo-complex, suggesting that HGF recognizes this loop in an induced-fit manner. A cluster of charged residues in the loop of c-MET, such as Glu302, Lys303, and Arg304, form multiple salt bridges with the charged residues in HGF, including Glu336, Asp338, and Arg373 (Fig. 2e). Notably, it has been long known that the maturation and functionality of c-MET are achieved by proteolytic cleavage between Arg307 and Ser308 within this long loop[29]. Based on our structural model, we hypothesized that proteolytic cleavage of this loop would lead to increased structural plasticity, particularly for residues 301–307, which may, in turn, facilitate its induced-fit binding to HGF. This structural analysis partially explains why proteolytic cleavage is critical for the activity of c-MET receptor. Next, we introduced multiple mutations in HGF, including point mutations (W321R, E336R, E361R, R373E, and Y376A), double mutation (W321R/Y376A), and triple mutation (W321R/E361R/Y376A), designed to weaken the interface III. Although most of the point mutants exhibited only slightly reduced activity, both of double and triple mutants showed significant deficiency in inducing c-MET activation (Fig. 2f and Supplementary Fig. 5). Consistently, mutations of Tyr369 and Phe373, two c-MET residues localized at interface III, to alanine affected c-MET activation by HGF (Fig. 2g and Supplementary Fig. 6). Moreover, the HGF W321R/E361R/Y376A mutant showed reduced ability in binding to c-MET in the pull-down experiment, further supporting our structural model (Fig. 2h, i).

Interface IV is mainly formed between the second and third blades of β-propeller in the SEMA domain of c-MET I and SPH domain of HGF I (Fig. 2b). This interface is nearly identical to that observed in the crystal structure of c-MET-SEMA and HGF-SPH complex[14], and will not be described in detail there. To confirm that interface IV is essential for c-MET activation, we introduced Y673A mutation into HGF to disrupt this interface, and showed that HGF Y673 mutant has a great deficiency in binding to c-MET as well as promoting c-MET activation (Fig. 2f, h, i and Supplementary Fig. 5).

**Interactions between c-MET II and HGF I**. Distinct to the binding mode between c-MET I and HGF I that involves multiple domains of HGF, only the K1 domain located on the opposite side of HGF I makes direct contact with c-MET II, indicating that HGF-K1 is essential for c-MET activation (Fig. 3a, b). In detail, two short loops of HGF I-K1 (residues 154–163; residues 192–192) make numerous contacts with a concave surface formed together by the fourth, fifth, and sixth blades of the c-MET II-SEMA, burying ~1141 Å² solvent-accessible surface area (Fig. 3c). Several charged residues from HGF I-K1, including His158, Glu159, Arg181, Glu195, Arg197, and Arg554, participate in this

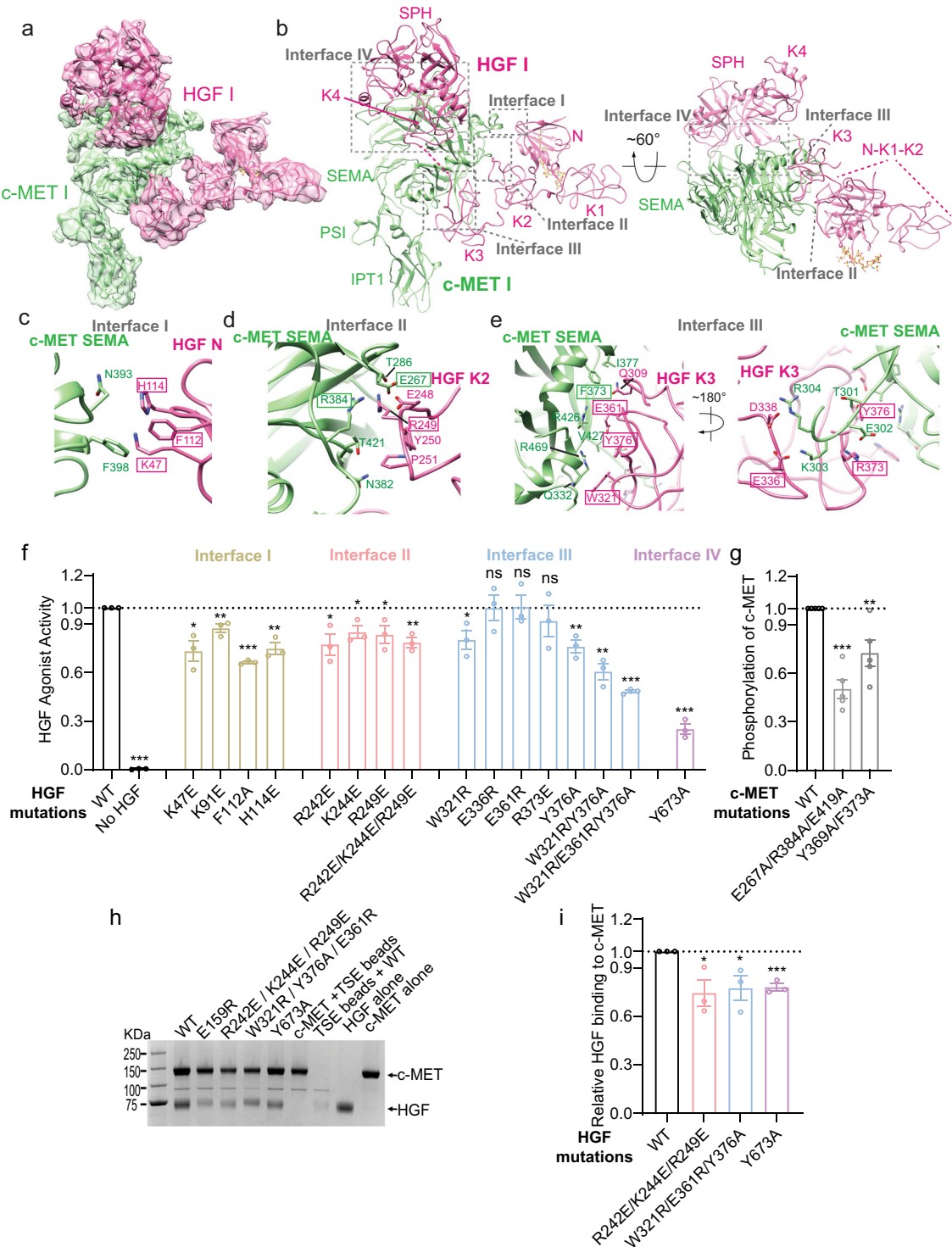

interaction. In addition, hydrophobic residues from HGF I-K1, such as MET155, Ile156, Pro157, and Pro194, pack tightly against residues Gly334, Tyr369, Phe373, and Ile377 in c-MET II-SEMA (Fig. 3c). Similar to the binding pattern between c-MET I-SEMA and HGF I-K3, the loop of c-MET (residues 300–311) that undergoes proteolytic cleavage during maturation also contributes to the binding between c-MET II-SEMA and HGF I-K1. After proteolysis cleavage, the newly formed C-terminus of c-MET α-chain inserts into a small pocket in HGF I-K1, which is mainly stabilized by the electrostatic interaction between Glu302 of c-MET and Arg197 of HGF (Fig. 3c). This observation further

indicates the functional significance of proteolytic cleavage in c-MET activation. Intriguingly, the residues of c-MET II-SEMA that are involved in the binding to HGF I-K1 largely overlap with those at the c-MET I-SEMA/HGF I-K3 interface, suggesting that c-MET-SEMA cannot simultaneously bind to HGF-K1 and K3. The implication of this structural observation will be discussed later.

To confirm the binding mode between c-MET II and HGF I as shown in our structure, we introduced three point mutations in the K1 domain of HGF, including E159R, E195R, and R197E, to disrupt this interface, and examined the effects of these mutations

**Fig. 2 Four interfaces between c-MET I and HGF I. a** 3D reconstruction of the 1:1 c-MET I/HGF I complex after focused 3D refinement of holo-complex, and the corresponding ribbon representation of this complex fitted into the cryo-EM map at 4.5 Å resolution. **b** The ribbon representation of the c-MET I/ HGF I complex shown in side and top views. Four distinct interfaces that are involved in the interaction between c-MET I and HGF I are indicated by dash boxes. Heparin is shown in yellow as a stick model. **c–e** Close-up views of interfaces I, II, and III. The HGF and c-MET residues mutated for cell-based activity assays are indicated by rectangular boxes. **f** The levels of c-MET autophosphorylation in response to HGF wild-type (WT) or indicated mutants that were expected to disrupt c-MET I/HGF I interaction. Mean ± SEM are from $N = 3$ independent biological repeats. The representative raw western blot data were shown in Supplementary Fig. 5. **g** HGF-induced c-MET autophosphorylation in H1299 cells expressing c-MET WT or indicated mutants that were expected to disrupt C-MET I/HGF I interaction. Mean ± SEM are from $N = 5$ independent biological repeats. The representative western blot data were shown in Supplementary Fig. 6. **h** SDS-PAGE of TSE pull-down binding assay. TSI tagged c-MET927 was immobilized to TSE resin as bait and incubated with HGF WT and various of mutations as prays. **i** Quantification of the pull-down binding results for the HGF R242E/K244E/R249E, W321R/E361R/ Y376A, and Y673A mutants shown in **h**. Mean ± SEM are from $N = 3$ independent biological repeats. Statistical difference in **f**, **g**, and **i** was analyzed using two-tailed Student's $t$-test, and $P$ values were calculated between WT and mutants: ns, $P > 0.05$; *$P \leq 0.05$; **$P \leq 0.01$; ***$P \leq 0.001$. Source data are provided as a Source Data file. Exact $P$ values in **f** from left to right: <0.0001, 0.0124, 0.0083, <0.0001, 0.0021, 0.0251, 0.0184, 0.0361, 0.0020, 0.0284, 0.9957, 0.9414, 0.4498, 0.0038, 0.0013, <0.0001, <0.0001. Exact $P$ values in **g** from left to right: <0.0001, 0.0042. Exact $P$ values in **i** from left to right: 0.0353, 0.0424, 0.0006.

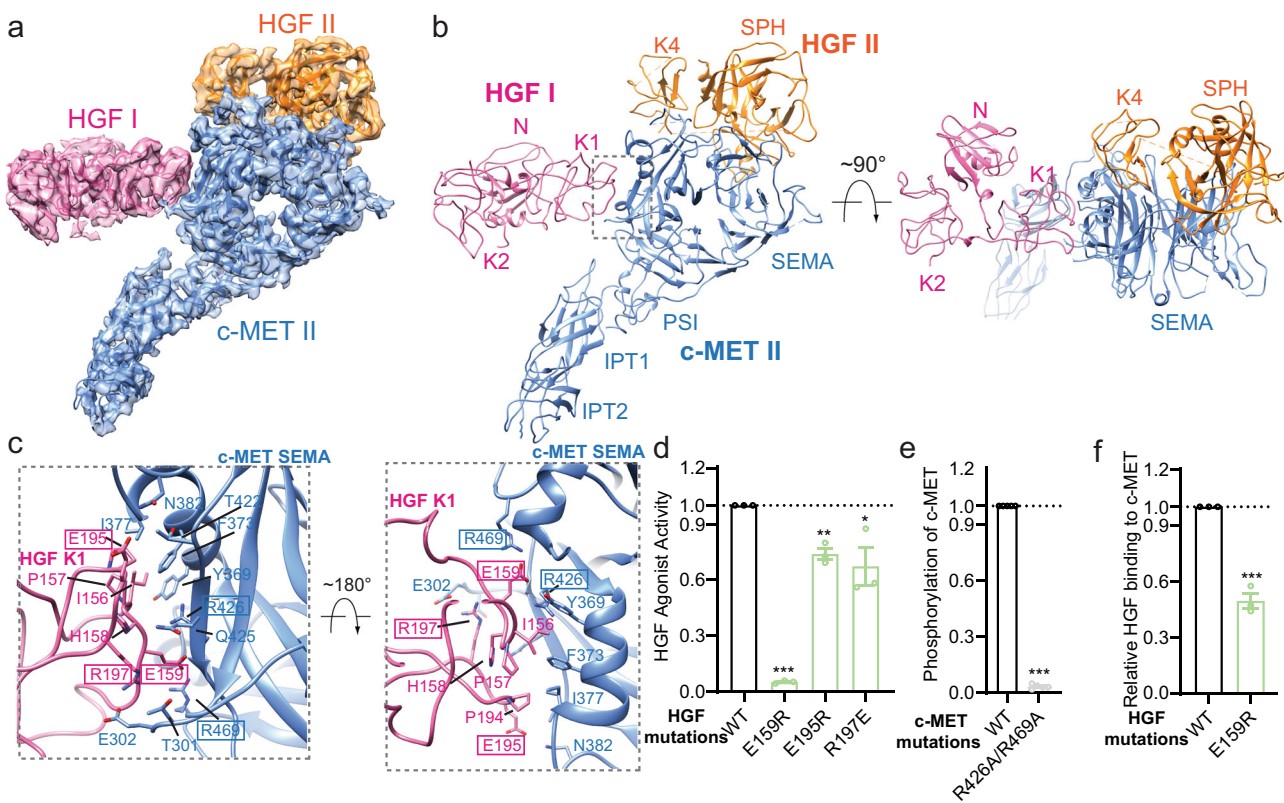

**Fig. 3 Interface between c-MET II and HGF I. a** 3D reconstruction of the c-MET II/HGF I/HGF II sub-complex and the corresponding ribbon representation of this complex fitted into a cryo-EM map of sub-complex at 4 Å resolution. **b** The ribbon representation of the c-MET II/HGF I/HGF II complex shown in side and top views. **c** Detailed view of the c-MET II-SEMA/HGF I-K1 interface shown in two different views. The HGF and c-MET residues mutated for cell-based activity assays are indicated by rectangular boxes. **d** The levels of c-MET autophosphorylation in response to HGF wild-type (WT) or indicated mutants that were expected to disrupt c-MET II/HGF I interaction. Mean ± SEM are from $N = 3$ independent biological repeats. The representative western blot data were shown in Supplementary Fig. 5. **e** HGF-induced c-MET autophosphorylation in H1299 cells expressing c-MET WT or indicated mutants that were expected to disrupt C-MET II/HGF I interaction. Mean ± SEM are from $N = 5$ independent biological repeats. The representative western blot data were shown in Supplementary Fig. 6. **f** Quantification of the pull-down binding result for the HGF E159R mutant shown in Fig. 2h. Mean ± SEM are from $N = 3$ independent biological repeats. Statistical difference in **d–f** was analyzed using two-tailed Student's $t$-test, and $P$ values were calculated between WT and mutants: ns, $P > 0.05$; *$P \leq 0.05$; **$P \leq 0.01$; ***$P \leq 0.001$. Source data are provided as a Source Data file. Exact $P$ values in **d** from left to right: <0.0001, 0.0012, 0.0319. Exact $P$ value in **e**: <0.0001. Exact $P$ value in **f**: 0.0002.

on c-MET activation. While E195R and R197E mutations reduced the biological activity of HGF to a certain level, the E159R mutation in HGF led to dramatically decreased phosphorylation of c-MET (Fig. 3d and Supplementary Fig. 5). Similarly, the R426A/R469A double mutant of c-MET cannot undergo autophosphorylation when stimulated with wild-type HGF (Fig. 3e and Supplementary Fig. 6). In consistent with the results

of the cell-based activity assay, our in vitro pull-down assay results showed that the HGF E159R mutant exhibited significantly lower binding affinity to the c-MET, as compared with wild-type HGF (Figs. 2h and 3f). These results together confirm that the c-MET II-HGF I interface as observed in the structure is important for the HGF-induced activation of c-MET.

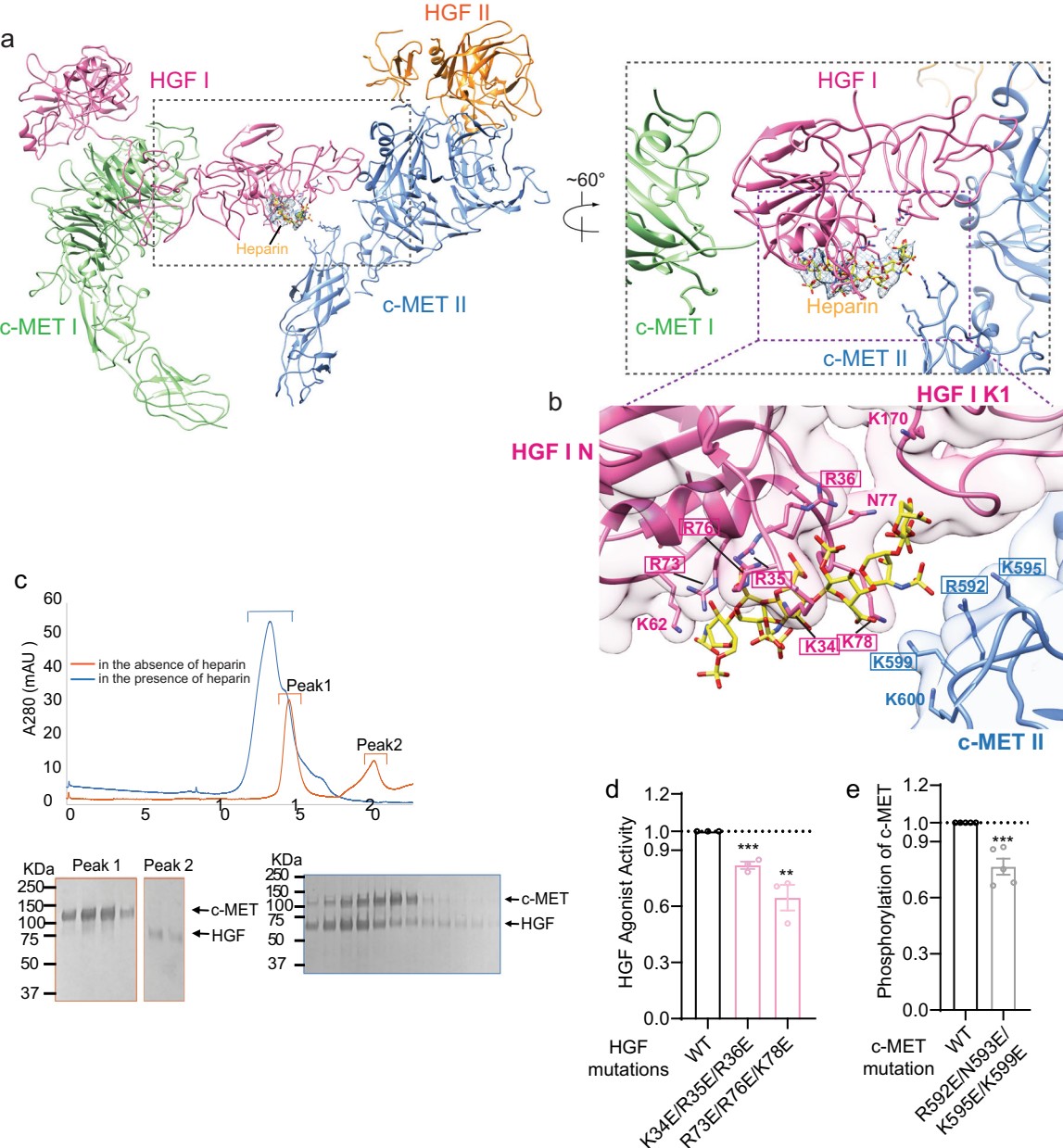

**Fig. 4 Heparin strengthens the interaction between c-MET II and HGF I. a** Overview of the binding of heparin at c-MET/HGF holo-complex, shown in two different views. The location of the heparin-binding site is between the IPT1 domain of c-MET II and N domain of HGF I. **b** Close-up view of the heparin-binding between the c-MET II-IPT1 domain and the HGF I-N domain. The HGF and c-MET residues mutated for cell-based activity assays are indicated by rectangular boxes. **c** The representative SEC profiles of c-MET/HGF complexes in the absence (orange) or presence (blue) of heparin from $N = 3$ repeats. The stable c-MET/HGF complex is formed only when heparin is present. **d** The levels of c-MET autophosphorylation in response to HGF wild-type (WT) or indicated mutants that were expected to disrupt the binding between HGF and heparin. Mean ± SEM are from $N = 3$ independent biological repeats. The representative western blot data were shown in Supplementary Fig. 5. **e** HGF-induced c-MET autophosphorylation in H1299 cells expressing c-MET WT or indicated mutants that were expected to disrupt c-MET/heparin interaction. Mean ± SEM are from $N = 5$ independent biological repeats. The representative western blot data were shown in Supplementary Fig. 6. Statistical difference in **d**–**e** is analyzed using two-tailed Student's $t$-test and $P$ values were calculated between WT and mutants: ns, $P > 0.05$; *$P \leq 0.05$; **$P \leq 0.01$; ***$P \leq 0.001$. Source data are provided as a Source Data file. Exact $P$ values in **d** from left to right: 0.0007, 0.0068. Exact $P$ value in **e**: 0.0006.

**c-MET II-HGF I-heparin interaction**. Our structure provides an explanation for the previous observation that heparin promotes HGF-induced c-MET activation[18]. Our cryo-EM map of holo-complex shows strong elongated density peaks between the IPT1 domain of c-MET II and N domain of HGF I that can be assigned to heparin, based on the size and shape of the density as well as the local chemical environment (Fig. 4a and Supplementary Fig. 2). A heparin polymer molecule with five repeating units of

disaccharides can be well fitted into cryo-EM density (Fig. 4a and Supplementary Fig. 2). However, the resolution of our cryo-EM map is insufficient to determine the sequence of heparin and the precise positions of each sulfate group.

Our model shows that the binding of heparin to the N domain of HGF I is coordinated by a cluster of positively charged residues, including Lys34, Arg35, Arg36, Lys60, Lys62, Arg73, Arg76, and Lys78 (Fig. 4b). Consistent with our model, residues

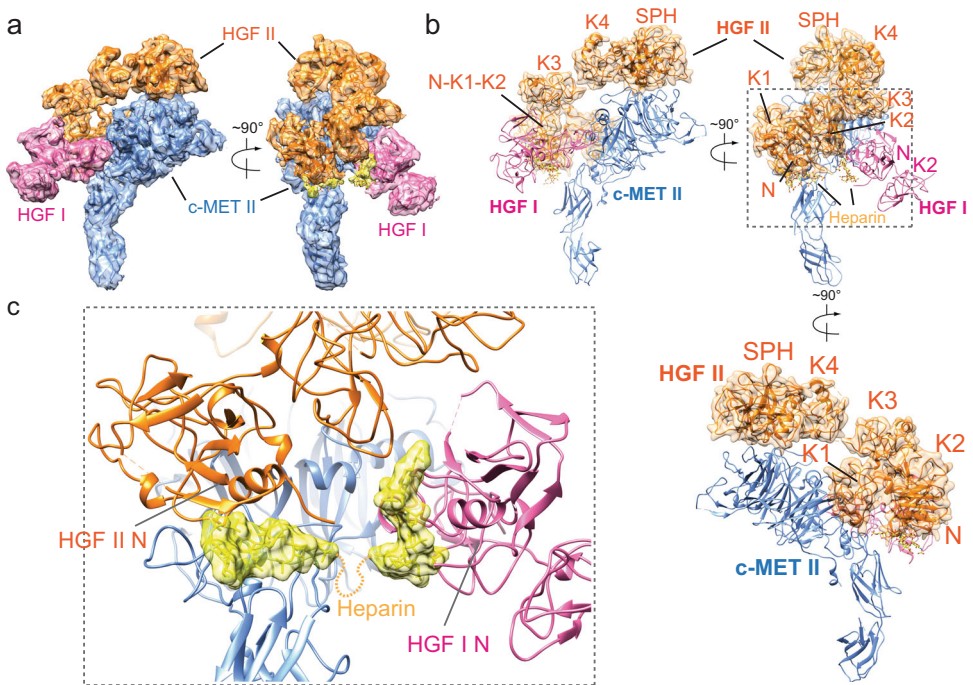

**Fig. 5 A rigid HGF II stabilizes the interaction between c-MET II and HGF I. a** 3D reconstruction of the c-MET II/HGF I bound with an HGF II with rigid conformation, and the corresponding ribbon representation of this complex fitted into the cryo-EM map at 4.9 Å resolution, shown in two orthogonal views. **b** The ribbon representation of c-MET II/HGF I in complex with a complete version of HGF II shown in two orthogonal views. The model of the intact HGF II is also shown as surface representation. **c** Detailed view of heparin-binding at the N domains of HGF I and II. A long heparin polymer could mediate the interaction between HGF I and II. The heparin molecules are shown as surface representation.

Lys58, Lys60, Lys62, Arg73, and Arg76 of HGF have been demonstrated previously by mutagenesis and NMR studies to be critical for heparin binding[19,20]. Importantly, the sulfate groups of heparin also make hydrogen bonds with several positively charged residues in an extended loop of c-MET II-IPT1, such as Arg592, Lys595, and Lys599 (Fig. 4b). Apparently, the concurrent binding of heparin to both c-MET II and HGF I contributes to the high-affinity association between c-MET II and HGF I; as such, our results provide a molecular basis for understanding how heparin promotes the activation of c-MET.

We first performed SEC to confirm the importance of heparin in the c-MET/HGF complex formation. In the absence of heparin, c-MET and HGF were eluted as two separate peaks in SEC (Fig. 4c). In the presence of heparin, however, c-MET and HGF were co-eluted as an earlier peak, indicating heparin significantly increases the binding affinity between c-MET and HGF (Fig. 4c). To further test the functional relevance of heparin-binding and validate our structural model, we introduced two triple mutations to the N domain of HGF (K34E/R35E/R36E and R73E/R76E/K78E) and tested their effect on c-MET activation. Both mutants showed markedly reduced HGF-dependent c-MET activation (Fig. 4d and Supplementary Fig. 5). We next introduced a quadruple mutation into c-MET-IPT1 (R592E/N593E/K595E/K599E), which was designed to fully disrupt the interaction between heparin and c-MET. Consistent with our structural model, the c-MET quadruple mutant showed reduced activation in response to HGF, as compared with wild-type c-MET (Fig. 4e and Supplementary Fig. 6).

**HGF II stabilizes c-MET II-HGF I interaction by forming a tripartite interface**. In the majority of the particles, the second HGF (i.e., HGF II) adopts a flexible conformation with only its SPH domain, which contacts c-MET II-SEMA, being well resolved in the cryo-EM map. Nevertheless, after further 3D classification, a cryo-EM map reconstructed from a smaller class of particles reveals a complete version of HGF II that forms a tripartite interaction with c-MET II and HGF I (Fig. 5a, b and Supplementary Fig. 3). The structures of the NK2 segment of HGF II can be superimposed well onto that of HGF I, suggesting that no domain rearrangements occur in this structural segment (Supplementary Fig. 4e). However, as compared with the structure of HGF I, the K3, K4, and SPH domains of HGF II undergo a large structural relocation with respect to NK2 (Supplementary Fig. 4e). This kind of conformational change is made possible due to the weak association between K3 and NK2 domains in HGF.

In the 1:2 c-MET/HGF complex, the SPH and K1 domains of HGF II contact the SEMA and PSI domains of c-MET, respectively; while the K2 and K3 domains of HGF II also make weak interaction with the N domain of HGF I (Fig. 5a, b). Additionally, two well-defined cryo-EM densities were identified in the N domains of both HGFs that could be attributed to heparins (Fig. 5c). These two heparin molecules are bound at the same site within the N domains of two HGFs and are surrounded by the same set of positively charged residues as described above. Notably, weak cryo-EM density was observed between two closely adjacent heparins, suggesting that the two heparin molecules are likely to be part of the same heparin polymer. Hence, a long heparin polymer could crosslink HGF I and II by simultaneously binding both HGFs and thereby stabilize the c-MET II-HGF I-HGF II tripartite interaction (Fig. 5c). It is worth emphasizing that, distinct to HGF I that contacts both c-MET I and II, HGF II only engages with c-MET II. This structural observation indicates that HGF I is the activating ligand that promotes c-MET dimerization and activation, whereas HGF II plays an auxiliary role that strengthens the interaction between HGF I and c-MET II.

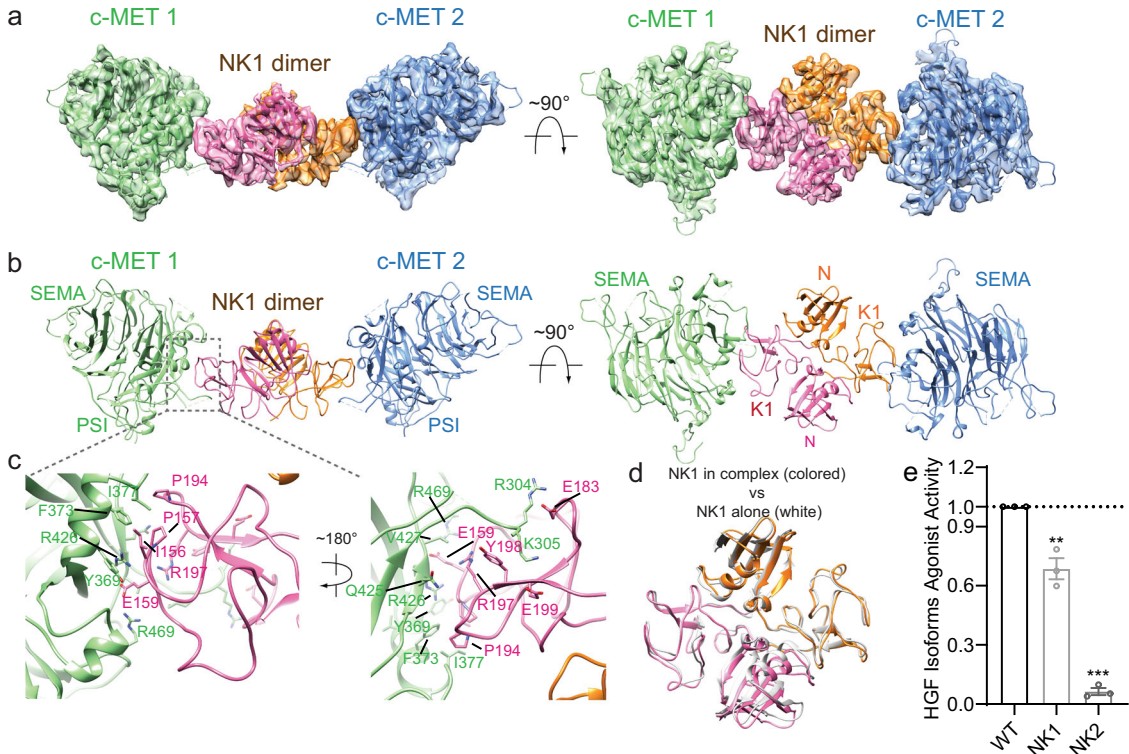

**Fig. 6 Overall structure of the c-MET/NK1 complex. a** 3D reconstruction of the 2:2 c-MET/NK1 complex, and the corresponding ribbon representation of this complex fitted into the cryo-EM map at 5 Å resolution, shown in two orthogonal views. **b** The ribbon representation of the 2:2 c-MET/NK1 complex shown in two orthogonal views. **c** Close-up view of the interface between c-MET-SEMA and HGF-K1, showing in two views. **d** Superposition between free NK1 dimer (PDB ID: 1NK1[44]; white) and NK1 dimer after binding c-MET (colored). **e** The levels of c-MET autophosphorylation in response to NK1 or NK2. Mean ± SEM are from $N = 3$ independent biological repeats. Statistical difference was analyzed using two-tailed Student's $t$-test and $P$ values were calculated between WT and mutants: **$P \leq 0.01$; ***$P \leq 0.001$. The representative western blot data were shown in Supplementary Fig. 5. Source data are provided as a Source Data file. Exact $P$ values in **e** from left to right: 0.0043, <0.0001.

**Structural model of c-MET/NK1 dimeric complex.** NK1 is the native isoform of HGF, acting as an alternative c-MET agonist[21]. To understand how NK1 activates c-MET, we determined the cryo-EM structure of the 2:2 c-MET/NK1 complex at an overall 5 Å resolution (Supplementary Figs. 1 and 7). Due to the structural flexibility, the entire stalk regions of two c-METs in the c-MET/ NK1 complex were poorly resolved that the IPT1 and IPT2 domains of c-MET can be only visualized in the cryo-EM map low-pass filtered by local resolutions (Supplementary Fig. 7f). Although the cryo-EM map was determined at medium resolution, we could build a reliable model by fitting the crystal structures of NK1 dimer as well as the SEMA, PSI, IPT1, and IPT2 domains of c-MET into a cryo-EM map with minor adjustment[24,30] (Supplementary Fig. 7). In contrast to the binding mode between c-MET and HGF, NK1 forms a stable head-to-tail dimer on its own, which recruits two c-METs to both sides in a symmetric manner (Fig. 6a, b). In such structural configuration, the two c-METs are placed in relatively close proximity, which enables their intracellular KDs to undergo autophosphorylation and thus initializing the downstream signaling (Fig. 6b and Supplementary Fig. 7f). Such ligand-induced dimerization model has been observed in the activation of c-MET by a bacterial c-MET agonist—InlB[31], as well as in the activation of many other RTKs, such as PDGFR and VEGFR[32,33].

NK1 engages with c-MET through the K1–SEMA interaction, which is virtually identical to that utilized in the formation of the c-MET II/HGF I complex (Fig. 6c). Hence, it will not be described in detail here. We superimposed the model of NK1 dimer onto that of free NK1 dimer, which was previously determined by X-

ray crystallography[30]. This comparison revealed no major structural differences between the free and c-MET bound NK1 (Fig. 6d). It has been proposed that heparin is able to promote the dimerization of NK1 as well as NK1 induced c-MET activation[30]. Indeed, weak densities that could be attributed to heparin were observed on top of two N domains of NK1 dimer, but cannot be modeled due to the low resolution. It is reasonable to hypothesize that NK2, another naturally occurring splicing variant of HGF, can bind c-MET via similar K1–SEMA interaction, but cannot activate c-MET due to the lack of capability for dimerization (Fig. 6e)[25].

## Discussion

Our cryo-EM and functional analyses indicate that the concurrent binding of one HGF to two c-METs by utilizing two completely distinct interfaces leads to c-MET dimerization and activation. This specific conformation of c-MET dimer is able to bring the two KDs into close proximity for *trans*-autophosphorylation. This 1:2 stoichiometry of HGF binding to c-MET is consistent with previous c-MET-HGF binding assays that indicated the coexistence of multiple c-MET binding sites at HGF[13]. The binding of HGF to c-MET I is mediated by the interaction between the SEMA of c-MET and the N, K2, K3, and SPH domains of HGF I. The binding of HGF to c-MET II is mainly driven by the interaction between the SEMA of c-MET and the K1 domain of HGF I. Heparin and a second HGF (HGF II) could further stabilize this interaction through the simultaneous binding to both HGF I and c-MET II (Fig. 7). Together, our study

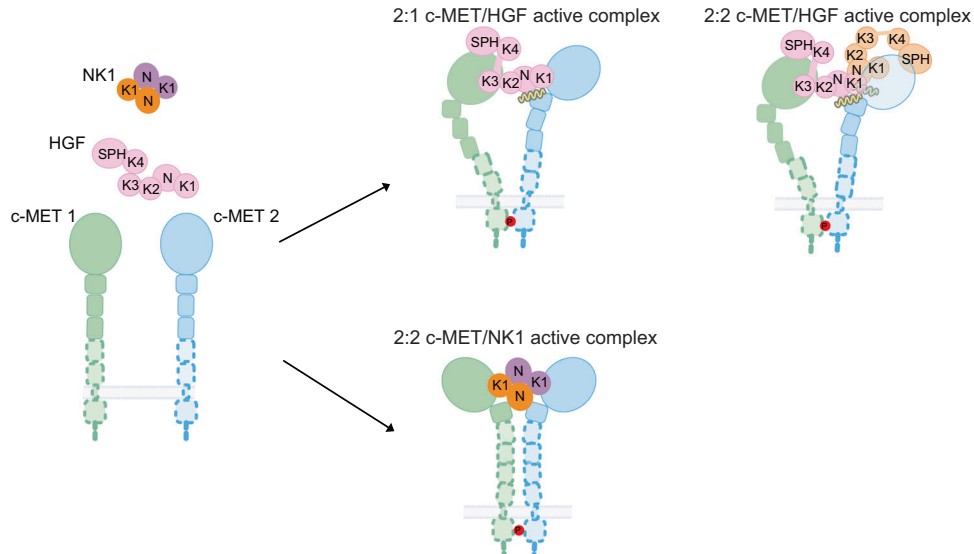

**Fig. 7 Cartoon representation of a working model for HGF and NK1 induced c-MET activation.** HGF induces an asymmetric c-MET signaling complex, whereas NK1 induces a symmetric c-MET signaling complex. The bindings of a second HGF and heparin strengthen the 2:1 c-MET/HGF active complex.

provides key mechanistic insight into the HGF-induced c-MET activation.

The SEMA of c-MET is the only domain involved in the direct interaction with HGF. In sharp contrast, 5 out of 6 domains of HGF I, including N, K1, K2, K3, and SPH domains, make direct contacts with either c-MET I-SEMA or c-MET II-SEMA, burying 61, 1141, 509, 665, and 962 Å$^2$ surface area, respectively. Moreover, our cell-based activity assays showed that the HGF with mutations that disrupt either K1–SEMA or SPH–SEMA interaction no longer activated c-MET, but the HGF with mutations that disrupt either N–SEMA, K2–SEMA, or K3–SEMA interaction still exhibited partial activity, suggesting that the K1–SEMA and SPH–SEMA interfaces are more critical for HGF-induced c-MET activation than any other interfaces.

Intriguingly, the two HGFs (HGF I and II) present in c-MET/ HGF holo-complex have completely distinct roles. HGF I is the activating ligand that plays a vital role in c-MET dimerization and activation. HGF II, however, only plays a supplementary role in c-MET activation. Our cryo-EM analysis shows that, in most of c-MET/HGF particles, HGF II adopts a flexible conformation with only the SPH domain touching the SEMA domain of c-MET II. The HGF II with a flexible N-K1-K2-K3 fragment might be able to recruit a third c-MET through the K1–SEMA interaction. In such a structural configuration, a number of c-METs and HGFs molecules could, in principle, assemble into higher-ordered oligomers in a linear arrangement on the cell membrane. The formation of higher-order clusters might be important for the activation and regulation of c-MET, like that proposed for several other RTKs[34].

In a small subset of particles, HGF II becomes rigid, and contacts both HGF I and c-MET II, with the help of a long heparin molecule. In this particular conformation, HGF II binding strengthens the interaction between HGF I and c-MET II and thereby stabilizes the entire c-MET I/HGF I/c-MET II complex (Fig. 7). As mentioned above, K1–SEMA and K3–SEMA interfaces are largely overlapped, meaning that once the SEMA of c-MET II is occupied by the K1 of HGF I with high affinity, the K3 of HGF II cannot bind to the same SEMA. This explains how the K3 of HGF II is dislocated from the SEMA of c-MET II, and instead contacts the N domain of HGF I, which then becomes the driving force for the dimerization of two HGFs. Moreover, although the surface of HGF II-K1 for c-MET binding is exposed,

the binding of another c-MET to HGF II-K1 in such rigid conformation would clash with HGF I bound c-MET II on the stalk region, explaining why rigid HGF II in the 2:2 complex cannot recruit additional c-MET molecule. It is tempting to hypothesize that this 2:2 c-MET/HGF complex may exhibit higher activity than the 2:1 complex, due to the high structural stability. Such a unique system may allow c-MET to respond differently at low or high HGF concentrations.

In this work, we provide the structural basis for the functional role of heparin-binding in HGF-induced c-MET activation. First, our model shows that a heparin molecule is sandwiched between the N domain of HGF I and the IPT1 domain of c-MET II, and acts as a "glue" to hold these two domains together. In this way, heparin contributes to the strong binding between HGF I and c-MET II. Furthermore, a long heparan molecule could potentially bind to both of HGF I and II simultaneously, thus stabilizing the 2:2 c-MET/HGF complex. It is also worth noting that a large cluster of long heparan polymers on the cell surface could be associated with multiple dimeric c-MET/HGF complexes and in principle induce the higher-order oligomerization of c-METs, and potentially lead to the formation of even larger biomolecular condensates on the cell membrane.

It has been long known that HGF, and its natural splicing variant NK1, are both able to activate c-MET, but HGF exhibits higher biological activity than NK1. We determined the cryo-EM structures of both c-MET/HGF and c-MET/NK1 complexes in the active state. The comparison of these two complexes reveals several major structural differences. First, HGF recruits two c-METs in an asymmetrical manner that contrasts with the C2 symmetry observed in c-MET/NK1 complex (Fig. 7). Secondly, the relative orientations and distance of two membrane-proximal regions are also significantly different between these two complexes, which to a certain extent could explain why HGF and NK1 have different ability in triggering c-MET activation. In addition to different signal strength, the distinct conformation of c-MET triggered by different types of ligand may allow it to preferentially activate different downstream signaling pathways through engaging distinct sets of downstream effectors, which is reminiscent of the "biased agonism" paradigm established for other RTKs, such as EGFR[35].

In conclusion, our structural and functional analyses provide a large body of novel information for defining the molecular

mechanism underlying the activation of c-MET in response to HGF and NK1 and paves the way for the eventual therapeutic intervention of diseases caused by aberrant activation of c-MET.

## Methods

**Constructs design**. All primers are listed in Supplementary Table 1.

*C-MET927*. Human cDNA encoding the c-MET extracellular region (amino acids 1–927) followed by the Human Rhinovirus 3 C (HR3C) recognition site, Tsi3 tag[36], and the 6-histidine-tag (His6) tag, was cloned into the pEZT-BM, which is a BacMan vector that can infect both Sf9 insect and HEK293F cells (Supplementary Table 1).

*C-MET927-LZ*. c-MET (amino acids 1–927) followed by a GGGGS linker, a GCN4 zipper sequence (RMKQL EDKVE ELLSK NYHLE NEVAR LKKLV GER), HR3C, Tsi3 tag, and His6 tag, was cloned into the pEZT-BM vector (Supplementary Table 1).

*HGF*. Full-length human HGF followed by HR3C site, and a His6 tag was cloned into the pEZT-BM vector (Supplementary Table 1).

All HGF mutants were generated using site-directed mutagenesis (NEB Q5 Site-Directed Mutagenesis). For the phosphorylation assay, a full-length c-MET receptor followed by five or two repeats of the Myc tag sequence was cloned into pLVX-IRES-Puro. To introduce mutations to c-MET, short gene fragments (100–150 nt) containing mutations were synthesized (IDT Inc.), and the corresponding gene fragment in the wild-type c-MET was replaced by the synthesized gene fragment using HiFi DNA assembly enzyme (NEB).

### Expression of c-MET, HGF, and NK1

*C-MET927, c-MET927-LZ, and HGF*. The plasmid was transformed into the DH10Bac bacteria (Geneva Biotech) for the production of bacmid DNA. Recombinant baculovirus was produced by transfecting Sf9 cells with the bacmid DNA using Cellfectin (Cellfectin Reagent Invitrogen). Protein was expressed with suspension-adapted HEK293F cells by infecting the virus at a ratio of 1:10 (virus: cell, v/v). The infected cells were supplemented with 5 mM sodium butyrate to increase protein expression. For HGF expression, 5% of FBS (Sigma) was supplemented to the culture media. The cells were cultured for 96 h at 37 °C and 8% $CO_2$.

*NK1*. Human HGF (amino acids 31–209) was tagged at the N-terminus with Thiol: disulfide interchange protein (DsBC) and His6 followed by HR3C recognition site cloned into pET22b vector. The plasmid was transformed into the Rosetta-gami 2 (DE3) (Milipore sigma) and cultured at 37 °C. When the OD600 reached 0.6, the cells induced with Isopropyl β-d-1-thiogalactopyranoside (IPTG), and expressed at 22 °C for 16 h.

### Protein purification

*C-MET927 and c-MET927-LZ purification*. Culture media containing the secreted protein of interest was harvested by centrifugation and filtered using a glass fiber filter. The cleared media was incubated with the INDIGO-Ni resin (Cube Biotech Inc.) for 30 min on the bottle roller. The resin was collected using a chromatography column. The resin was washed with 10 resin volumes of the washing buffer (20 mM Hepes pH 7.5, 400 mM NaCl, 20 mM Imidazole) and eluted with the elution buffer (20 mM Hepes pH 7.5, 400 mM NaCl, 400 mM Imidazole). The eluted protein was dialyzed against the dialysis buffer (20 mM Hepes pH 7.5, 400 mM NaCl) at 4 °C for 16 h. Tsi3 and His6 tags were cleaved with an HR3C protease during the dialysis. The protein was further purified by using SEC (Superose 6 increase 100/30 Cytiva) with the SEC buffer (20 mM Hepes at pH 7.5 and 400 mM NaCl).

*HGF purification*. Culture media containing the secreted HGF was harvested by centrifugation and filtered using a glass fiber filter. The cleared media was adjusted to a final concentration of 50 mM sodium/potassium phosphate at pH 6.5 and applied to the cation exchange column (Hitrap SP HP, Cytiva). The protein was eluted with a linear gradient over 50% buffer B (20 mM Sodium/potassium phosphate buffer pH 6.5 and 2 M NaCl). The eluate was further purified using SEC (Superose 6 increase 100/30 Cytiva) with the SEC buffer (20 mM Hepes pH7.5 and 400 mM NaCl).

*NK1 purification*. Cells were harvested and lysed by sonication in sonication buffer (20 mM Hepes pH 7.5, 400 mM NaCl, 5% glycerol). After centrifugation, the supernatant was applied to the Ni Sepharose resin (Cytiva). The DsBC and His6 tag were cleaved by adding HR3C protease to the resin and incubating for 16 h at 4 °C. Eluted protein was further purified with SEC (Superdex 75, Cytiva) in the SEC buffer (20 mM Hepes pH 7.5 and 400 mM NaCl).

**C-MET927-LZ/HGF/heparin and c-MET927-LZ/NK1/heparin complex reconstitution**. c-MET927-LZ was mixed with HGF or NK1 in the presence of heparin (Tinzaparin sodium; MilliporeSigma) in a 1:1:4 molecular ratio. The complex sample was concentrated and loaded onto SEC (SRT SEC-500 Sepax) in the buffer containing 20 mM Hepes pH 7.5 and 100 mM NaCl. The fractions containing the complex were pooled and concentrated using an Amicon Ultra concentrator with a 100 kDa cut-off (Millipore) to ~5 mg/ml.

**EM data acquisition**. Immediately before preparing the cryo-EM grids, 2 mM fluorinated Fos-Choline-8 (Anatrace) was added to the purified sample to prevent the protein from being damaged by the hydrophobic air–water interface on a cryo-EM grid. The cryo-EM grid was prepared by applying 3 μl of the protein samples to glow-discharged Quantifoil R1.2/1.3 300-mesh gold holey carbon grids (Quantifoil, Micro Tools GmbH, Germany). Grids were blotted for 4.0 s under 100% humidity at 4 °C before being plunged into the liquid ethane using a Mark IV Vitrobot (FEI). Micrographs were acquired on a Titan Krios microscope (FEI) operated at 300 kV with a K3 direct electron detector (Gatan), using a slit width of 20 eV on a GIF-Quantum energy filter. SerialEM was used for the data collection[37]. A calibrated magnification of 46,296 was used for imaging, yielding a pixel size of 1.08 Å on images. The defocus range was set from −1.6 to −2.6 μm. Each micrograph was dose-fractionated to 30 frames under a dose rate of 18 e⁻/pixel/s, with a total exposure time of 4 s, resulting in a total dose of about 60 e⁻/Å².

**Image processing**. Movie frames of c-MET927-LZ/HGF micrographs were motion-corrected and binned twofold, resulting in a pixel size of 1.08 Å, and dose-weighted using MotionCor2[38]. CTF correction was performed using GCTF[39]. The rest of the image processing steps were carried out using RELION 3.1[40]. Particles were first roughly picked by using the Laplacian-of-Gaussian blob method and then subjected to 2D classification. Class averages representing projections of the c-MET/HGF in different orientations were used as templates for reference-based particle picking. A total of 3,287,093 particles were picked from 8100 micrographs. Particles were extracted and binned three times (leading to 3.24 Å/pixel) and subjected to another round of 2D classification. Particles in good 2D classes were chosen (1,367,618 in total) for 3D classification using an initial model generated from a subset of the particles in RELION. After the initial 3D classification c-MET927-LZ/HGF particles set, two major classes were identified showing good secondary structural features (namely holo-complex and sub-complex, respectively). The holo-complex comprises two c-MET and two HGF molecules; while the sub-complex only comprises one c-MET and two HGF. Particles for holo-complex or sub-complex were selected separately and re-extracted into the original pixel size of 1.08 Å. Subsequently, we performed finer 3D classification for each particle subset by using local search in combination with small angular sampling, resulting in improved density for both assembly states. The final reconstructions of holo-complex and sub-complex were resolved at 4.8 and 4 Å resolution, respectively. To improve the resolution for c-MET I and HGF I in holo-complex, we performed focused 3D refinement, by using a soft mask around c-MET I and HGF I, leading to a 3D reconstruction at an overall resolution of 4.5 Å. Finally, we performed another round of finer 3D classification on the particle set of sub-complex. After the further classification, we could identify a new class where HGF II adopts a rigid conformation. The subsequent 3D refinement of this class yielded a cryo-EM at 4.9 Å resolution, resolving a complete version of HGF II.

A total of 1,342,144 particles were picked from 3644 micrographs of c-MET/NK1. A total of 656,258 particles were selected by 2D classification. Subsequent 3D classification revealed one good class showing good density. Next, we performed two rounds of finer 3D classification by using local search in combination with small angular sampling, resulting in a new class showing improved density for the entire protein. 3D refinement of this class, along with CTF refinement and particle polishing, yielded a structure at 5 Å resolution. The resolution was estimated by applying a soft mask around the protein density. The Fourier Shell Correlation (FSC) 0.143 criterion was used. Local resolution was calculated in RELION.

**Model building, refinement, and validation**. Model buildings of both c-MET/HGF and c-MET/NK1 complexes were initiated by rigid-body docking of individual domains from the crystal structures of SEMA, PSI, IPT1, and IPT2 domains of c-MET and N, K1, K2, and SPH domains of HGF[14,24,25,30]. The models of K3 and K4 of HGF were first built by homology modeling using the structures of K1 of HGF as a template (50% sequence identity between K1 and K3; 36.1% sequence identity between K1 and K3), and then rigid-body fitted into the cryo-EM density in Chimera[41]. The manual building was carried out using the program Coot. The model was refined by using the real-space refinement module in the Phenix package (V1.17)[42]. Restraints on secondary structure, backbone Ramachandran angels, and residue sidechain rotamers were used during the refinement to improve the geometry of the model. MolProbity 4.5 as a part of the Phenix validation tools was used for model validation (Table 1). Figures were generated in Chimera 1.14.

**c-MET activation assay**. Naive H1299 cells were gifted from John Minna (UT Southwestern Medical Center, Dallas) and cultured in RPMI 1640 medium (Thermofisher) supplemented with 10% (vol/vol) FBS at 37 °C and 5% $CO_2$. Lentiviruses encoding myc-tagged c-MET (myc-c-MET, WT, or mutant cloned

into the pLVx-IRES-puro vector) were generated in HEK 293 T packaging cells following standard transfection protocol[43]. Titrations of lentiviruses were used to infect H1299 cells, and infected cells were selected using 2 µg/ml puromycin for 2–3 days. Cells stably express near-endogenous myc-c-MET (WT or mutant) were selected for the following assays.

To examine the agonist activities of HGF mutants, naive H1299 cells were plated 200,000 cells/well on six-well plates (Corning) and starved in an FBS-free medium for 16 h. The starved cells were stimulated with 2 ml, 100 ng/ml HGF (WT or mutant) for 10 min at 37 °C. After stimulation, cells were washed once with 1x PBS and then collected in 150 µl 2x Laemmli Sample Buffer supplemented with 5% 2-Mercaptoethanol. SDS-PAGE and Western Blotting were used to probe c-MET (8198 S, Cell Signaling, 1:1000 dilution in 5% milk), phospho-c-MET (Tyr1234/1235, 3077 S, Cell Signaling, 1:1000 dilution in 5% milk), and loading control β-actin (A1978, Sigma, 1:5000 dilution in 5%BSA). Blotting results were analyzed using ImageJ.

To examine the phosphorylation capacities of c-MET mutants, 3′UTR siRNA (hs.Ri.MET.13.1 from IDT) was used for selective knockdown of endogenous c-MET in H1299 cell lines that stably express near-endogenous myc-c-MET (WT or mutant). Transfections of siRNA were carried out using Lipofectamine RNAi-Max (Life Technologies) following the manufacturer's protocol. After depletion of endogenous c-MET, 100 ng/ml HGF WT was used to stimulate the phosphorylation of myc-c-MET following the protocol described above.

**Pull-down assay**. Five micrograms of the bait protein c-MET927-TSI3-His was immobilized to TSE resin by incubating for 30 min on ice and washed with binding buffer PBS (Gibco) supplemented with10 µM Tinzaparin and 2 mM CaCl2. The resin was incubated with 5 µg of the prey proteins HGF wild type or mutants in 50 µl of binding buffer for 1 h at 4°. Each tube was tap gently but thoroughly every 10 min. The resin was washed with binding buffer to wash away prays. The resin was boiled in SDS loading buffer (Bio-rad) and loaded on the SDS-PAGE. The intensity of the bands was calculated using ImageJ.

**Reporting Summary**. Further information on research design is available in the Nature Research Reporting Summary linked to this article.

## Data availability
We have deposited all the five maps/models into the PDB/EMDB database, including 2:2 holo-complex (PDB: 7MO7; EMD-23919), c-MET I/HGF I complex (PDB: 7MO8; EMD-23920), c-MET II/HGF I/HGF II (only contains K4 and SPH) (PDB: 7MO9; EMD-23921), c-MET/HGF I/intact HGF II (PDB: 7MOA, EMD-23922), and c-MET/NK1 (7MOB; EMD-23923). Source data are provided with this paper.

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

## Acknowledgements

Cryo-EM data were collected at the University of Texas Southwestern Medical Center (UTSW) Cryo-Electron Microscopy Facility, funded in part by the Cancer Prevention and Research Institute of Texas (CPRIT) Core Facility Support Award RP170644. We thank Dr. Stoddard for facility access. This work is supported in part by grants from the National Institutes of Health (R35GM130289 to X.Z and R01GM143158 to X.-C.B.), the Welch Foundation (I-1702 to X.Z. and I-1944 to X.-C.B.), and CPRIT (RP160082 to X.-C.B.). X.-C.B. and X.Z. are Virginia Murchison Linthicum Scholars in Medical Research at UTSW. Z.C. is supported by a grant from the National Institutes of Health (GM73165 to Marcel B. Mettlen).

## Author contributions

X.Z. and X.-C.B. conceived the project. E.U. performed the protein expression and complex formation procedures, prepared the cryo-EM samples, and performed the pull-down assays. X.-C.B. and X.Z. determined the cryo-EM structures and built the models. Z.C. did the cell-based c-MET activation assays, with the help from G.-Y.X., X.Z., and X.-C.B. wrote the paper with inputs from other authors.

## Competing interests

The authors declare no competing interests.
