## [Peer Review File · Nature Communications]

REVIEWER COMMENTS

Reviewer #1 (Remarks to the Author):

The authors determined by cryo-EM the long-awaited structures of the hepatocyte growth factor (HGF) receptor c-MET dimerized by HGF and by the natural HGF splice variant NK1. The mode of activation is completely different for both ligands. A single HGF bridges two c-MET molecules in an asymmetric complex, while dimeric NK1 forms a C2 symmetric complex with two c-METs. This manuscript describes an experimental tour de force. The results are a milestone in understanding how HGF and its natural splice variant NK1 activate c-MET. The work confirms that both HGF and c-MET are structurally highly flexible proteins making them very difficult targets for structure determination. The authors used a clever strategy to tame this flexibility to at least some degree by adding a leucine zipper dimerization motif to the C-terminus of the c-MET extracellular region. Despite of forced c-MET dimerization, large portions of the C-terminal c-MET extracellular region are not resolved in the density.

The structures are at relatively low resolution, but they are consistent with each other. Extensive and well controlled receptor activation assays with HGF or c-MET variants carrying point mutations in various binding interfaces support the correctness of these structures. The results are by and large in agreement with previous data from other groups, including low resolution crystallographic c-MET/HGF and c-MET/NK1 structures mentioned in a review by Blaszczyk et al. (*Progress in Biophysics and Molecular Biology*, 2015), which have not yet been published as original work.

I have some general questions. Maybe the authors could add short comments on this in the revised version of their manuscript:

1) Figure 5: Is the structure of the NK2 fragment of the rigid HGF II the same as that of the extended HGF I and the crystal structure described by Tolbert (PNAS 2010)?

2) The authors speculate that tethering of the N-terminal segment to the stalk region may stabilize c-MET in a specific extended conformation (lines 177-178). What about the c-MET/NK1 structure: Is the N-terminus of c-MET tethered via a disulfide to IPT1? If so, how much flexibility does this allow? Can the flexibility that is still possible with tethering the N-terminus to IPT1 explain IPT1 being unresolved in this structure? Would IPT1 (and maybe IPT2) become visible, if no C2 symmetry was imposed during image processing?

3) c-MET is highly glycosylated. Is there density for sugars? Did the authors model them? Does any of the interfaces contain a predicted N-glycosylation site?

4) The authors argue that no higher-order complexes can be formed because binding of a third c-MET to the K1 domain of HGF II in the holo-complex with a complete HGF II would lead to clashes. However, the N-K1-K2-K3 fragment of HGF II in the holo-complex is very flexible, as mentioned e.g. in lines 128-130. Thus, one could imagine that the N-K1-K2-K3 fragment of HGF II does not bind c-

MET II but instead adopts a different conformation that allows another c-MET to bind the K1 domain of HGF II. Actually, this does not seem unlikely to me, because the K1/SEMA interface buries the largest surface area and is probably the interaction with highest affinity. Binding of a third c-MET to HGF II would lead to chain-like clusters. Could the more than 95% discarded particle images contain such higher order structures?

I have some major issues that should be addressed

1) Line 108-110: For the low-resolution structure of non-dimerized c-MET, the authors mention the C-terminal c-MET domains being in close proximity. Which domains do they refer to? Would that be IPT2 as in their higher resolution structures or IPT4 as the real C-terminal domain of the c-MET extracellular region? How close are the C-termini? Can this distance be bridged by the short GGGGS linkers connecting to the GCN4 dimerization domain?

2) The authors apparently will not deposit a structure of the holo complex (2:2 c-MET:HGF complex), at least they did not provide the corresponding PDB validation report. I'm puzzled by this fact. For the general reader, this would be the most valuable structure. Is there a reason, why this most important structure will not be deposited, although it is often referenced in the text and also shown in Fig. 1 and Fig. 4?

3) Roughly 98% of particles after 2D classification were not used for the final 3D reconstruction. Could they contain other, maybe functionally relevant, conformations of the c-MET/HGF or the c-MET/NK1 complex? Is it normal to keep such a tiny fraction of 2D classes?

4) Figure 7: I consider this model speculative and only of limited value. The authors do not seem to present structural evidence for their two middle states (Partially active state, Full active state). Instead, their EM structure with HGF II visible only from K4 (N-K1-K2-K3 invisible) is not shown here. Do the authors have any evidence for the indicated order of binding events, i.e. first binding of SPH to c-MET I and K1 to c-MET II, later binding of K3 to c-MET I and N+heparin to c-MET II? The depicted transition from Partially active state to Full active state implies a movement of HGF I SPH domain on c-MET I Sema domain and substantial conformational changes in the stalk region of both c-MET molecules. The first is certainly wrong, and there seems to be no experimental data backing the second. In contrast to the model, the existence of the subcomplex (c-MET II / HGF I / HGF II) suggests to me that first HGF I and HGF II bind to c-MET II and subsequently HGF I binds c-MET I.

5) As in Fig. 7, the abstract suggests the initial formation of a 2:1 c-MET/HGF complex, to which a second HGF can bind. The cryo-EM data instead seem to suggest the initial formation of a 1:2 c-MET/HGF complex to which a second c-MET can bind.

6) The authors state that they will deposit the cryo-EM map and the coordinates into EMDB and PDB upon acceptance. This order is not the community-agreed standard any more. Structure and map deposition should be a prerequisite for acceptance of the paper.

Minor issues:

Line 57: “transcription factors” instead of “transcription”

Line 136 and 143: The authors mention “entire c-MET” and “a complete model for the entire 2:2 c-MET-LZ/HGF complex”. However, IPT3 and IPT4 seem to be missing in both cases. Maybe this could be phrased more accurately.

Line 240: The authors mention “the high affinity binding between c-MET I-SEMA and HGF I-K3”, but do not provide binding data or an actual affinity. I suggest re-phrasing the sentence accordingly.

Line 252: “Y376A” instead of “Y375A”

Line 262 (2x) and Fig. 2F: The mutation T673A in HGF is mentioned. However, residue 673 in HGF is a tyrosine. In Extended Data Figure 5 this is given correctly as Y673A.

Line 288: “R426A” instead of “R456A”

Line 320: “c-MET quadruple mutant showed greatly reduced activation”. This seems exaggerated to me.

Line 351-353: Such ligand induced dimerization in a C2 symmetric complex was also described for the 2:2 c-MET:InlB complex (Ferraris et al. 2010).

Line 420: “c-MET/NK1” instead of “HGF/NK1”

Line 740: “HGF I-N domain” instead of “HGF II-N domain”

Figure 2B: The authors could mention in the legend that heparin is shown as stick model with yellow carbons.

Figure 2C-E: It would be nice to show inter-molecular hydrogen bonds and salt bridges.

Figure 2G: The R426A/R469A mutant probes the c-MET II / HGF I interface and would more logically fit to Fig. 3. If the authors want to keep it in Fig. 2, they should mention this fact in the figure legend.

Figure 4C: Which c-MET construct was used for SEC? Monomeric or dimerized c-MET927? Is this a non-reducing gel? If it is reducing, two bands should be visible for both HGF and c-MET.

Figure 4C: Were equimolar amounts of HGF and c-MET loaded onto the SEC column? The gel suggests that the bulk of HGF elutes earlier than the bulk of c-MET. Could this be indicative for formation of 1:2 c-MET:HGF complexes eluting early and some free c-MET eluting late?

Figure 5: Two heparin molecules are shown, but only one ligand (most likely heparin) is mentioned in the corresponding PDB validation report entitled “PDB_reprot_cMETI_HGFI_IntactHGFI”.

Line 753: “c-MET II/HGF I” instead of “c-MET I/HGF I”

Figure 7: The model does not show the interaction between N or K2 and c-MET I (interfaces I and II).

Extended Data Fig. 1: Are these non-reducing gels? If they are reducing, two bands should be visible for both HGF and c-MET.

There are two “Extended Data Fig. 7”, while “Extended Data Fig. 6” is missing.

(First) Extended Data Fig. 7: The R592E/N593E/K595E/K599E variant runs at a lower molecular weight than all other variants. What is the reason? Is it the number of myc tags?

(First) Extended Data Fig. 7: The E267A/R384A/E419A variant appears to have an unusually large fraction of unprocessed single-chain c-MET. Do the authors have any idea why?

(Second) Extended Data Fig 7c: The cryo-EM map of c-MET/NK1 complex colored by local resolution seems to contain IPT1, maybe also IPT2. However, the final model lacks both IPT1 and IPT2. Is the density sufficient, to model their position at least roughly? If so, is their conformation similar to that in c-MET I and c-MET II of the holo complex with HGF?

Table 1: For the structure “c-MET II / HGF I / HGF II (K4, SPH)” 1 heparin is mentioned as ligand. However, the corresponding PDB validation report entitled “PDB_reprot_cMETI_HGFI_PartHGFI” shows that no ligand is present.

Minor / formatting

Line 131: “were absent” instead of “were absence”

Line 144, 146, 163 and 523: “holo-complex” instead of “homo-complex”

Line 154: “mediated” instead of “mediate”

Line 210: “decreased” instead of “deceased”

Line 218: “positively charged” instead of “positive charged”

Line 219: “negatively charged” instead of “negative charged”

Line 268: “contacts” instead of “contact”

Line 276: “after proteolytic cleavage” instead of “after proteolytic”

Lines 297, 408, 411: “molecule” instead of “molecular”

Line 302: “charged” instead of “changed”

Line 334: “heparin molecules” instead of “heparins molecules”

Lines 340, 395, 736: “strengthens” instead of “strengths”

Line 342: “isoform” instead of “isoforms”

Line 361: “hypothesize” instead of “hypothesis”

Line 394: “contacts” instead of “contact”

Line 397: “mentioned” instead of “mention”

Line 421: “extent” instead of “extend”

Line 462: "Protein purification" instead of "Proteins purification"

Lines 480/481: "20 mM" instead of "20 Mm"

Line 489: "fractions" instead of "factions"

Line 562: "phospho-c-MET" instead of "phosphor-c-MET"

Line 777: "in both samples" instead of "in both sample"

Line 810: "Phospho-c-MET" instead of "Phospho-c-Mer"

Line 819: "expressing" instead of "express"

Reviewer #2 (Remarks to the Author):

In this article the authors present two structural studies by cryo-EM and single particle analysis of the c-MET receptor, a member of the receptor tyrosine kinase (RTK) family, in complex with two of its activating ligands, hepatocyte growth factor (HGF) and NK1. The results, together with a series of mutagenesis study within in vivo c-MET activation assays, allow a clear description of the interactions between the different partners and also highlight the crucial role of heparin in this process. Based on these findings, the authors suggest a mechanism model of c-MET activation upon HGF binding. This study brings new insights into the activation mechanism of the c-MET receptor, the latter playing essential roles in many aspects of the cell physiology, adding new hints to the knowledge on RTKs activation biology.

Major comments:

- The cryo-EM work is of great quality, from sample preparation to processing and model building, representing the state-of-the-art of the current workflows. Despite a limited resolution of 4 to 5 Å, the authors perfectly described the entire complexes and interactions, and were even able to validate and confirm their findings at the side chain level by using mutagenesis studies and activation assays. All the results and interpretations are solid and do not suffer from any "overinterpretation" that can be a danger at these resolutions. The figure are quite clear and support nicely the descriptions of the complexes in the text.

- I am more concerned by the activation model proposed by the authors as I have the feeling that it would need some more biochemical experiments to clearly validate (or refute) their statements. More specifically, the model suggests that one HGF molecule binds to a c-MET dimer (or dimerise two c-MET molecules), followed by the stabilisation of the complex upon the binding of a second HGF molecule. I can't see in the paper any references clearly supporting this particular order of events nor results. One could also think that HGF binds to all c-MET molecules via SPH-K4-K3 domains for HGF and SEMA domain for c-MET, and then in a second step only dimerisation occurs with K1 domain binding to a second cMET-HGF complex, displacing some of the HGF domains. Within the manuscripts the authors often refer to "high affinity", "low affinity", "weak interaction" without providing any actual measurement; having some numbers/measurements would allow a real comparison of the different binding phenomenons described in the complex. Since all components are purified in vivo, I would suggest the authors to also perform binding experiments using classical biophysics technics in order to characterise better the formation of the complex and the c-MET dimerisation as it is a key step of the activation. Figure 7, representing the model, needs also to be clearer and better described with a legend. I am still not sure to understand on what the "partially active state" is based on. I would also integrate a model for NK1 activation!

Minor comments:

- line 62: "multiple disulphide binds"; how many? Two are visible on fig.1.

- line 93, 119...: "active state": since the construct used doesn't have a TM domain nor a kinase domain, I do think it is a dangerous statement; some rephrasing might be needed, maybe by using "mimicking" or similar words.

- line 111: "leucine zipper motif"; the GNC4 zipper sequence trick is used for decades now in the RTK field, a reference is needed.

- line 163: "homo-complex" should be "holo-complex".

- line 210: "deceased" should be "decreased".

- line 211: first introduction fo the activation assays with various mutants; please introduce a bit more the experiment to make it clear how it was done.

- line 257: Extended Fig.6 doesn't exist, there are two Extended Fig.7 in the documents available for the review.

- line 261: "interface I" should be "interface IV".

- Figure 1b: I would remove the "grey membrane" as the construct used lacks TM and kinase domains. I would replace it with a "cartoon" zipper.

- Figure 4: The panels need to be rearranged; the current reading order is a, d, b and c.

- Cryo-EM data table: there is an extra "/" between "c-MET" and "II" on the third column of data.

Felix Weis

Reviewer #3 (Remarks to the Author):

This manuscript by Uchikawa et al. addresses a long-standing question of how HGF and related ligands are able to activate the c-MET receptor tyrosine kinase. This is a very relevant question given the multiple oncogenic roles c-MET plays in human cancers. The authors use cryo-EM to explore the stoichiometry of ligand binding, the composition of the ligand-receptor interfaces, the role of heparin sulphate (a surrogate GAG) and differences between members of the same ligand family in binding c-MET. This study reveals how a single HGF can dimerise two c-MET receptors, while a second HGF molecule plays an auxiliary role in strengthening the receptor dimer. The close proximity of two heparan sulphate GAG binding sites on HGF-1/HGF2 and c-MET II suggests HS stabilises the c-MET dimer conformation. There is an excellent comparison between HGF and NK1 ligands, the former ligand drives an asymmetric 2:2 complex while the latter stabilises a symmetric C2 2:2: complex. The study is convincing and technically sound with clear and well-presented figures. Overall, it's a very compelling story with novelty and the authors are to be congratulated on

answering a long-standing question. It is likely to drive future work to explore differences in NK versus HGF driven signalling through c-MET. I would therefore recommend this study for publication. I have gathered together some typos in the manuscript below and raise some questions and minor points that may improve the manuscript clarity.

Minor points

1. A few typos I noticed;

Line 108 – features

Line 131 – absent

Line 144 – holo-complex? Also elsewhere in the manuscript, line 146 etc.

Line 208 – interacts "with"

Line 210 – decreased

Line 228 – deficiency

Line 276 – proteolysis

Line 286 – to a certain level

Line 332 – by the same set

Line 385 – unstable

Line 395 – strengthens

Line 405 – Such a unique

Line 408 – heparin molecule

Line 411 – heparan molecule could potentially

Line 413 – heparan polymers

Line 421 – which to a certain extent could explain

Line 736 – Heparin strengthens

Line 804 – may be better to say "(d) Modular structure of of HGF"

2. Heparin sulphate (Tinzaparin sodium - an anticoagulant) is used as a surrogate for heparan sulphate GAG presented by proteoglycans. It may be good to mention this at some point in case the casual reader thinks it is heparin that participates in c-MET activation.

3. Figure 2 and 3 improvements – it would help if the location of the HGF or c-MET mutations were emphasised in Figures 2 and 3 for clarity, either a box around the residues mutated or underlined would help. I would indicate that panel 2f is targeting HGF residues, whereas Figure 2g panel targets c-MET residues. I would indicate which interface residues belong to in panel 2g.

4. In view of this study it may (or may not) be interesting to comment on a reinterpretation of the high and low affinity binding sites for HGF originally found for cells transfected with c-MET, proposed to be the high affinity receptor with heparin/heparan as the low affinity receptor.

5. It would be helpful to mention where the pivot point is for c-MET stalk region conformational change is located in Extended Data Figure 4? Is it within a domain or within a connecting linker region?

6. It would be good to clarify that pEZT-BM is a Bacmam vector that can be used to infect both HEFK293F cells as well as Sf9 insect cells.

7. It may be helpful to clarify in line 822, what is meant by actin1 and actin2 used for the normalisation of c-MET phosphorylation.

We thank the reviewers for the positive and constructive comments. In the revised manuscript, we have addressed all the reviewers' concerns by doing additional experiments and by rewriting the manuscript. Doing so has significantly improved our manuscript.

Our point-by-point responses are listed below. For ease of reading, we have colored our responses in blue.

Reviewer #1 (Remarks to the Author):

The authors determined by cryo-EM the long-awaited structures of the hepatocyte growth factor (HGF) receptor c-MET dimerized by HGF and by the natural HGF splice variant NK1. The mode of activation is completely different for both ligands. A single HGF bridges two c-MET molecules in an asymmetric complex, while dimeric NK1 forms a C2 symmetric complex with two c-METs. This manuscript describes an experimental tour de force. The results are a milestone in understanding how HGF and its natural splice variant NK1 activate c-MET. The work confirms that both HGF and c-MET are structurally highly flexible proteins making them very difficult targets for structure determination. The authors used a clever strategy to tame this flexibility to at least some degree by adding a leucine zipper dimerization motif to the C-terminus of the c-MET extracellular region. Despite of forced c-MET dimerization, large portions of the C-terminal c-MET extracellular region are not resolved in the density. The structures are at relatively low resolution, but they are consistent with each other. Extensive and well controlled receptor activation assays with HGF or c-MET variants carrying point mutations in various binding interfaces support the correctness of these structures. The results are by and large in agreement with previous data from other groups, including low resolution crystallographic c-MET/HGF and c-MET/NK1 structures mentioned in a review by Blaszczyk et al. (Progress in Biophysics and Molecular Biology, 2015), which have not yet been published as original work.

We thank the reviewer for the positive assessment of our manuscript, and we appreciate the constructive comments which we have addressed below.

I have some general questions. Maybe the authors could add short comments on this in the revised version of their manuscript:

1) Figure 5: Is the structure of the NK2 fragment of the rigid HGF II the same as that of the extended HGF I and the crystal structure described by Tolbert (PNAS 2010)?

Thanks the reviewer for raising this good point. Yes, the structures of the NK2 fragment in HGFI and rigid HGFII are very similar. Both of them are also similar to the structure of NK2 alone determined previously by X-ray crystallography (Tolbert, PNAS 2010). The structural rigidity of NK2 is mainly due to the extensive interaction among N, K1 and K2 domains. In contrast, as compared with the structure of HGFI, the K3, K4 and SPH domains of the rigid HGFII undergo a large relocation with respect to the NK2. We have prepared a new figure panel e in Extended Data Figure 4, and added a few sentences in the main text to address this point.

2) The authors speculate that tethering of the N-terminal segment to the stalk region may stabilize c-MET in a specific extended conformation (lines 177-178). What about the c-MET/NK1 structure: Is the N-terminus of c-MET tethered via a disulfide to IPT1? If so, how much flexibility does this allow? Can the flexibility that is still possible with tethering the N-terminus to IPT1 explain IPT1 being unresolved in this structure? Would IPT1 (and maybe IPT2) become visible, if no C2 symmetry was imposed during image processing?

Thanks the reviewer for raising this good point. Although the stalk region of c-MET in c-MET/NK1 structure is less well resolved than that in c-MET/HGF structure, it could be clearly visualized in the low-pass filtered map as shown in Extended Data Figure 7c. In addition, the model of c-MET SEMA-PSI-IPT1-IPT2 derived from the structure of c-MET/HGF complex could be perfectly rigid-body fit into the low-pass filtered cryo-EM map of c-MET/NK1, as shown in our newly prepared Extended Data Figure 7f. This result suggests that (1) c-MET adopts very similar conformation when it is bound by either HGF or NK1; (2) the tethering of N-terminus to IPT1 indeed largely restricts the conformational flexibility of IPT1 and IPT2 of c-MET. We do notice that the distance between two IPT2 domains is much shorter in the structure of c-MET/HGF, as compared to that in the structure of c-MET/NK1. We speculate that certain degree of interaction between the stalk regions of two c-METs may exist in c-MET/HGF complex, but not in c-MET/NK1. Such interaction could further stabilize the conformation of stalks in the structure of c-MET/HGF complex, which partially explains why the stalk regions of c-MET in the c-MET/HGF complex were resolved at higher resolution. As indicated in Extended Data Figure 7d, the first round of 3D classification was performed without imposing any symmetry; however, one of the resulting classes showed a clear two-fold symmetry. Therefore, it is unlikely that the poorly resolved stalk regions of c-MET in the structure of c-MET/NK1 complex is due to the imposed C2 symmetry.

3) c-MET is highly glycosylated. Is there density for sugars? Did the authors model them? Does any of the interfaces contain a predicted N-glycosylation site?

There are totally 13 predicted N-glycosylation sites at c-MET. We could observe weak sugar density adjacent to asparagin residue for most of these N-glycosylation sites, among which the sugar densities for ASN45 and ASN202 sites are relatively well resolved. Still, we can't model them precisely, due to the low-resolution feature. We instead show the un-modelled sugar densities for ASN45 and ASN202 in Extended Data Figure 2. Based on our model, none of these N-glycosylations is involved in protein-protein interactions.

4) The authors argue that no higher-order complexes can be formed because binding of a third c-MET to the K1 domain of HGF II in the holo-complex with a complete HGF II would lead to clashes. However, the N-K1-K2-K3 fragment of HGF II in the holo-complex is very flexible, as mentioned e.g. in lines 128-130. Thus, one could imagine that the N-K1-K2-K3 fragment of HGF II does not bind c-MET II but instead adopts a different conformation that allows another c-MET to bind the K1 domain of HGF II. Actually, this does not seem unlikely to me, because the K1/SEMA interface buries the largest surface area and is probably the interaction with highest affinity. Binding of a third c-MET to HGF II would lead to chain-like clusters. Could the more than 95% discarded particle images contain such higher order structures?

Thanks the reviewer for making this good point. We completely agree with the reviewer that the HGF II with a flexible N-K1-K2-K3 fragment is able to recruit a third c-MET. In such manner, c-MET and HGF can together assemble into higher ordered oligomer in a linear arrangement. Supporting this hypothesis, we do observe some large particle aggregation in the electron micrographs, which may represents the high ordered oligomeric form of c-MET/HGF complex. Nevertheless, the structure of high-ordered c-MET/HGF complex cannot be captured in a relatively stable conformation, even after extensive 3D classification, probably due to the flexible linker between K2 and K3 domains. Still, such higher ordered linear oligomers are likely to be formed on the cell membrane, and they may become more stable on the cell membrane as their conformational plasticity may be partially restricted by the membrane. We have added more discussion in the main text to address this point.

I have some major issues that should be addressed

1) Line 108-110: For the low-resolution structure of non-dimerized c-MET, the authors mention the C-terminal c-MET domains being in close proximity. Which domains do they refer to? Would that be IPT2 as in their higher resolution structures or IPT4 as the real C-terminal domain of the c-MET extracellular region? How close are the C-termini? Can this distance be bridged by the short GGGGS linkers connecting to the GCN4 dimerization domain?

We apologize for the in clarity. The two IPT2 domains of non-dimerized c-MET/HGF are in proximity in our low-resolution cryo-EM structure. Given the flexibility of IPT3 and IPT4 relative to IPT2, the C-termini of the two c-MET molecules in principle could be placed right next to each other. We have prepared a new figure to show the fitting of c-MET/HGF model into the low-resolution cryo-EM map of non-dimerized c-MET/HGF complex. As shown in the figure below, the model of 2:1 c-MET/HGF complex reported in this work (c-MET I: green, c-MET II: blue, HGF: purple) can be fit well into the cryo-EM map of non-dimerized c-MET/HGF complex (colored in white), indicating that dimerized or non-dimerized c-MET/HGF complexes adopt very similar conformations. The only difference between these two structures is that the HGF II is not bound at c-MET II in the structure of non-dimerized c-MET/HGF complex, presumably because less HGF is added into cryo-EM sample of non-dimerized c-MET/HGF complex (2:1 c-MET/HGF molar ratio for non-dimerized c-MET sample versus 2:2 molar ratio used in this work). In the cryo-EM structure of non-dimerized c-MET/HGF complex, the IPT3 and IPT4 domains are also completely invisible, suggesting that IPT3 and IPT4 of c-MET are flexible in both samples. Due to the flexibility of IPT3 and IPT4 of c-MET, the dimerized GCN4 localized in the C-terminus of IPT4 domain is unlikely to introduce any structural restraints on the c-MET/HGF complex.

2) The authors apparently will not deposit a structure of the holo complex (2:2 c-MET:HGF complex), at least they did not provide the corresponding PDB validation report. I'm puzzled by this fact. For the general reader, this would be the most valuable structure. Is there a reason, why this most important structure will not be deposited, although it is often referenced in the text and also shown in Fig. 1 and Fig. 4?

We apologize for the confusion caused. The reason why we only provided the PDB validation reports for the cryo-EM structures of sub-complexes in the initial submission is because we only use the two cryo-EM maps of c-MET/HGF sub-complexes that were determined at high-resolution for the model building. The model for the entire 2:2 c-MET/HGF holo-complex was originally made by rigid-body fitting the models of sub-complexes into the cryo-EM map. During the revision, we have carried out additional refinement that included positional refinements of the atomic coordinates in Phenix. We have now deposited all the 5 maps/models, including 2:2 holo-complex (PDB: 7MO7; EMD-23939), c-MET I/HGF I complex (PDB: 7MO8; EMD-23920), c-MET II/HGF I/HGF II (only contains K4 and SPH) (PDB: 7MO9; EMD-23921), c-MET/HGFI/intact HGFII (PDB: 7MOA; EMD-23922) and c-MET/NK1 (7MOB; EMD-23923). We will also submit all the pdb reports for all these 5 models along with the revised manuscript.

3) Roughly 98% of particles after 2D classification were not used for the final 3D reconstruction. Could they contain other, maybe functionally relevant, conformations of the c-MET/HGF or the c-MET/NK1 complex? Is it normal to keep such a tiny fraction of 2D classes?

Thanks the reviewer for raising this issue. The cryo-EM data processing for both of c-MET/HGF and c-MET/NK1 samples was highly challenging due to high levels of particle heterogeneity. As shown in the data processing workflows for both datasets, the initial rounds of 3D classification eliminated "bad" particles that could not be correctly aligned, and separated particles with large conformational differences (holo-complex versus sub-complex in the case of c-MET/HGF sample). The subsequent rounds of 3D classification with finer angular sampling were carried out to obtain the highest possible resolution and map quality. Thus it is clear that all the remaining particles beyond the initial rounds of 3D classification have the same global conformation (Extended Data Figure 3 and 7), but further 3D classification and refinement were needed to select better preserved particles (less damage by air-water interface, etc) and obtain the best possible maps. A similar process has been used for other challenging cryo-EM structures, such as our previous work on gamma-secretase and STING (PMID: 26623517 and PMID: 30842659). We believe that the final refined maps reported in the manuscript represent most, if not all, the functionally relevant conformations that could be captured with the current datasets.

4) Figure 7: I consider this model speculative and only of limited value. The authors do not seem to present structural evidence for their two middle states (Partially active state, Full active state). Instead, their EM structure with HGF II visible only from K4 (N-K1-K2-K3 invisible) is not shown here. Do the authors have any evidence for the indicated order of binding events, i.e. first binding of SPH to c-MET I and K1 to c-MET II, later binding of K3 to c-MET I and N+heparin to c-MET II? The depicted transition from Partially active state to Full active state implies a movement of HGF I SPH domain on c-MET I Sema domain and substantial conformational changes in the stalk region of both c-MET molecules. The first is

certainly wrong, and there seems to be no experimental data backing the second. In contrast to the model, the existence of the subcomplex (c-MET II / HGF I / HGF II) suggests to me that first HGF I and HGF II bind to c-MET II and subsequently HGF I binds c-MET I.

Thanks the reviewer for the critical comments. We agree with the reviewer that we don't have experimental evidence to support the proposed order of binding events. It is also very speculative to claim the partially active state as it is not captured from our dataset. Therefore, we have removed the text in our original manuscript that speculated the order of binding events, and we have eliminated any claims that is related to the partially active state from the text and figure 7.

5) As in Fig. 7, the abstract suggests the initial formation of a 2:1 c-MET/HGF complex, to which a second HGF can bind. The cryo-EM data instead seem to suggest the initial formation of a 1:2 c-MET/HGF complex to which a second c-MET can bind.

Thanks the reviewer for making this good point. As mentioned in the response to point 1, if the molar ratio between c-MET and HGF is 2:1 in the cryo-EM sample, we could only capture the structure of 2:1 c-MET/HGF complex. This result suggests that how the complex is formed may also depends on the concentration of HGF relative to c-MET. Further studies are required to fully address this issue. To prevent misleading the readers, we have removed all the speculation related to the sequential formation of c-MET/HGF complex in the text as well as in figure 7.

6) The authors state that they will deposit the cryo-EM map and the coordinates into EMDDB and PDB upon acceptance. This order is not the community-agreed standard any more. Structure and map deposition should be a prerequisite for acceptance of the paper.

As mentioned in the response to point 2, we have now deposited all the 5 maps/models described in the manuscript to PDB/EEMDB database. All the maps/models will be released before publication.

Minor issues:

Line 57: "transcription factors" instead of "transcription"

Corrected.

Line 136 and 143: The authors mention "entire c-MET" and "a complete model for the entire 2:2 c-MET-LZ/HGF complex". However, IPT3 and IPT4 seem to be missing in both cases. Maybe this could be phrased more accurately.

Point accepted. We have phrased these two sentences to "majority part of c-MET extracellular domain (all domains except for IPT3 and IPT4)" and "a model for the majority part of 2:2 c-MET-LZ/HGF complex".

Line 240: The authors mention "the high affinity binding between c-MET I-SEMA and HGF I-K3", but do not provide binding data or an actual affinity. I suggest re-phrasing the sentence accordingly.

Point accepted. We have removed the "high affinity" from this sentence.

Line 252: "Y376A" instead of "Y375A"

Corrected.

Line 262 (2x) and Fig. 2F: The mutation T673A in HGF is mentioned. However, residue 673 in HGF is a tyrosine. In Extended Data Figure 5 this is given correctly as Y673A.

Corrected in both text and figure.

Line 288: “R426A” instead of “R456A”

Corrected.

Line 320: “c-MET quadruple mutant showed greatly reduced activation”. This seems exaggerated to me.

Point accepted. We have removed “greatly”.

Line 351-353: Such ligand induced dimerization in a C2 symmetric complex was also described for the 2:2 c-MET:InlB complex (Ferraris et al. 2010).

Good point. We have added a sentence in the text – “Such ligand induced dimerization model has been observed in the activation of c-MET by a bacterial c-MET agonist – InlB” and cited this paper in the revised in manuscript.

Line 420: “c-MET/NK1” instead of “HGF/NK1”

Corrected.

Line 740: “HGF I-N domain” instead of “HGF II-N domain”

Corrected.

Figure 2B: The authors could mention in the legend that heparin is shown as stick model with yellow carbons.

Good point. We have added “Heparin is shown in yellow as stick model”.

Figure 2C-E: It would be nice to show inter-molecular hydrogen bonds and salt bridges.

We attempted to show all the hydrogen bonds and salt bridges in Figure 2c-e, but the figures became very crowded after displaying them, as there are many hydrogen bonds and salt bridges between molecules. I think the figures will be clearer if we do not show the hydrogen bonds and salt bridges. The readers can download our models from PDB if you want to know more detailed information about the inter-molecular interactions.

Figure 2G: The R426A/R469A mutant probes the c-MET II / HGF I interface and would more logically fit to Fig. 3. If the authors want to keep it in Fig. 2, they should mention this fact in the figure legend.

Good point. We have moved the result of the R426A/R469A mutant to Fig. 3.

Figure 4C: Which c-MET construct was used for SEC? Monomeric or dimerized c-MET927? Is this a non-reducing gel? If it is reducing, two bands should be visible for both HGF and c-MET.

The construct used for this experiment is monomeric c-MET927 (not linked by GCN4 dimer). The gel shown in Figure 4C is the non-reducing gel. When the c-Met and HGF is reduced using DTT, three bands for c-MET (uncleaved band, beta- and alpha-subunits), two band for HGF (beta- and alpha-subunits). We attached here the SDS page results of c-MET and HGF under both of the non-reducing and reducing conditions.

Figure 4C: Were equimolar amounts of HGF and c-MET loaded onto the SEC column? The gel suggests that the bulk of HGF elutes earlier than the bulk of c-MET. Could this be indicative for formation of 1:2 c-MET:HGF complexes eluting early and some free c-MET eluting late?

20 μ M of c-MET and 20 μ M of HGF were present in the sample “c-MET HGF without Tinzaparin”; while 20 μ M of c-MET, 20 μ M of HGF and 80 μ M Tinzaparin were present in the sample “c-MET HGF with Tinzaparin”. The mixtures were injected to the size exclusion chromatography (superose 6 increase 10/300) in the buffer containing 20 mM Hepes pH 7.5, 200 mM NaCl.

We are aware of two populations in the SEC peak for the sample with Tinzaparin. This could indicate the formation of 1:2 c-MET:HGF complex, as suggested by the reviewer. However, we did not perform cryo-EM analysis for this peak separately.

Figure 5: Two heparin molecules are shown, but only one ligand (most likely heparin) is mentioned in the corresponding PDB validation report entitled “PDB_reprot_cMETI_HGFI_IntactHGFII”.

Thanks the reviewer for pointing out this issue. There is indeed an error in the original model of c-MET I/HGF I/Intact HGF II complex. We have corrected this modelling error. In the revised PDF report for the structure of c-MET I/HGF I/Intact HGF II complex, B and C chains correspond to heparin 1 and 2, respectively.

Line 753: “c-MET II/HGF I” instead of “c-MET I/HGF I”

Corrected.

Figure 7: The model does not show the interaction between N or K2 and c-MET I (interfaces I and II).

Thanks the reviewer for raising this issue. We have revised our activation model shown in figure 7. The revised cartoon model can better reflect the active structure of c-MET/HGF complex.

Extended Data Fig. 1: Are these non-reducing gels? If they are reducing, two bands should be visible for both HGF and c-MET.

These are non-reducing gels.

There are two “Extended Data Fig. 7”, while “Extended Data Fig. 6” is missing.

We apologize for the error. The figure number has been changed.

(First) Extended Data Fig. 7: The R592E/N593E/K595E/K599E variant runs at a lower molecular weight than all other variants. What is the reason? Is it the number of myc tags?

Yes, the R592E/N593E/K595E/K599E variant has two myc tags, while the others contains five myc tags. This has been mentioned in the Methods section.

(First) Extended Data Fig. 7: The E267A/R384A/E419A variant appears to have an unusually large fraction of unprocessed single-chain c-MET. Do the authors have any idea why?

We speculate that this variant might partially affect the accessibility of the furin cleavage site and thus increase the fraction of unprocessed single-chain c-MET.

(Second) Extended Data Fig 7c: The cryo-EM map of c-MET/NK1 complex colored by local resolution seems to contain IPT1, maybe also IPT2. However, the final model lacks both IPT1 and IPT2. Is the density sufficient, to model their position at least roughly? If so, is their conformation similar to that in c-MET I and c-MET II of the holo complex with HGF?

This point has been addressed in the response to the second general question. The IP1 and IPT2 domain of c-MET in the cryo-EM structure of c-MET/NK1 complex can only be visualized in the low-pass filtered map as shown in Extended Data Figure 7c, f. We didn't include the models of IPT1 and IPT2 in the structure of c-MET/NK1 complex, as the resolution for this part is too low for precise and confident model building. Based on the rigid-body modelling result, c-MET adopts very similar conformation when it is bound by either HGF or NK1.

Table 1: For the structure "c-MET II / HGF I / HGF II (K4, SPH)" 1 heparin is mentioned as ligand. However, the corresponding PDB validation report entitled "PDB_reprot_cMETI_HGFI_PartHGFI" shows that no ligand is present.

Thanks the reviewer for raising this issue. This error has been corrected in the revised model.

Minor / formatting

Line 131: "were absent" instead of "were absence"

Corrected.

Line 144, 146, 163 and 523: "holo-complex" instead of "homo-complex"

Corrected.

Line 154: "mediated" instead of "mediate"

Corrected

Line 210: "decreased" instead of "deceased"

Corrected

Line 218: "positively charged" instead of "positive charged"

Corrected

Line 219: "negatively charged" instead of "negative charged"

Corrected

Line 268: "contacts" instead of "contact"

Corrected

Line 276: "after proteolytic cleavage" instead of "after proteolytic"

Corrected

Lines 297, 408, 411: "molecule" instead of "molecular"

Corrected

Line 302: "charged" instead of "changed"

Corrected

Line 334: “heparin molecules” instead of “heparins molecules”

Corrected

Lines 340, 395, 736: “strengthens” instead of “strengths”

Corrected

Line 342: “isoform” instead of “isoforms”

Corrected

Line 361: “hypothesize” instead of “hypothesis”

Corrected

Line 394: “contacts” instead of “contact”

Corrected

Line 397: “mentioned” instead of “mention”

Corrected

Line 421: “extent” instead of “extend”

Corrected

Line 462: “Protein purification” instead of “Proteins purification”

Corrected

Lines 480/481: “20 mM” instead of “20 Mm”

Corrected

Line 489: “fractions” instead of “factions”

Corrected

Line 562: “phospho-c-MET” instead of “phosphor-c-MET”

phosphor

Line 777: “in both samples” instead of “in both sample”

Corrected

Line 810: “Phospho-c-MET” instead of “Phospho-c-Mer”

Corrected

Line 819: “expressing” instead of “express”

Corrected

Reviewer #2 (Remarks to the Author):

In this article the authors present two structural studies by cryo-EM and single particle analysis of the c-MET receptor, a member of the receptor tyrosine kinase (RTK) family, in complex with two of its activating ligands, hepatocyte growth factor (HGF) and NK1. The results, together with a series of mutagenesis study within in vivo c-MET activation assays, allow a clear description of the interactions

between the different partners and also highlight the crucial role of heparin in this process. Based on these findings, the authors suggest a mechanism model of c-MET activation upon HGF binding. This study brings new insights into the activation mechanism of the c-MET receptor, the latter playing essential roles in many aspects of the cell physiology, adding new hints to the knowledge on RTKs activation biology.

We thank Dr. Weis for these constructive comments that have led us to modify the manuscript as detailed below.

Major comments:

- The cryo-EM work is of great quality, from sample preparation to processing and model building, representing the state-of-the-art of the current workflows. Despite a limited resolution of 4 to 5 Å, the authors perfectly described the entire complexes and interactions, and were even able to validate and confirm their findings at the side chain level by using mutagenesis studies and activation assays. All the results and interpretations are solid and do not suffer from any “overinterpretation” that can be a danger at these resolutions. The figure are quite clear and support nicely the descriptions of the complexes in the text.

We thank Dr. Weis for the positive comments

- I am more concerned by the activation model proposed by the authors as I have the feeling that it would need some more biochemical experiments to clearly validate (or refute) their statements. More specifically, the model suggests that one HGF molecule binds to a c-MET dimer (or dimerise two c-MET molecules), followed by the stabilisation of the complex upon the binding of a second HGF molecule. I can't see in the paper any references clearly supporting this particular order of events nor results. One could also think that HGF binds to all c-MET molecules via SPH-K4-K3 domains for HGF and SEMA domain for c-MET, and then in a second step only dimerisation occurs with K1 domain binding to a second cMET-HGF complex, displacing some of the HGF domains. Within the manuscripts the authors often refer to “high affinity”, “low affinity”, “weak interaction” without providing any actual measurement; having some numbers/measurements would allow a real comparison of the different binding phenomenons described in the complex. Since all components are purified in vivo, I would suggest the authors to also perform binding experiments using classical biophysics technics in order to characterise better the formation of the complex and the c-MET dimerisation as it is a key step of the activation. Figure 7, representing the model, needs also to be clearer and better described with a legend. I am still not sure to understand on what the “partially active state” is based on. I would also integrate a model for NK1 activation!

We thank Dr. Weis for the critical comments. Reviewer 1 also raises the similar criticisms that our activation model is speculative and our proposed order of binding events lacks of experimental support. We agree with these points and have completely removed the text related to the speculation on the order of binding events. We also remade figure 7 in the revised manuscript. In the new figure 7, the model for the partially active state is removed, and the activation model for NK1 is added.

As suggested by Dr. Weis, we first attempted to test the binding between c-MET and HGF by using classic biophysical methods such as ITC and SPR. However, the low protein yield and the complexity of the interactions make it difficult to obtain binding affinities of individual binding sites with ITC or SPR. We instead used pull-down experiment to characterize the binding between c-MET and HGF.

Particularly, we designed four HGF mutations based on our structure, which we predicted would perturb K1-SEMA, K2-SEMA, K3-SEMA and SPH-SEMA interactions, respectively. We successfully purified these four HGF mutants, and used pull-down assays to test their ability of binding to c-MET. As a control, we also tested the binding of wild type HGF to c-MET with pull-down assay. As compared with wild type HGF, all the 4 HGF mutants exhibited various degrees of weaker affinity in binding to c-MET. Among these 4 mutations, E159R, which disrupts the K1-SEMA interaction, has the strongest effect in disrupting c-MET/HGF interaction, in consistent with our cell-based result showing that E159R mutation almost abolishes the HGF induced c-MET activation. Together, our new pull-down binding results further validate our cryo-EM model of c-MET/HGF complex. These new data has been described in the main text and presented in figure 2 and 3.

Minor comments:

- line 62: “multiple disulphide binds”; how many? Two are visible on fig.1.

To our best knowledge, we don't know for certain how many disulfide bonds there are between α - and β -subunits. However, 3 disulfide bonds can be observed in our model. We have changed the figure 1 and our text based on this structural observation.

- line 93, 119...: “active state”: since the construct used doesn't have a TM domain nor a kinase domain, I do think it is a dangerous statement; some rephrasing might be needed, maybe by using “mimicking” or similar words.

Pointed accepted. We have changed from “in the active state” to “mimicking the active state”.

- line 111: “leucine zipper motif”; the GNC4 zipper sequence trick is used for decades now in the RTK field, a reference is needed.

Pointed accepted. We have cited one paper that uses GNC4 to stabilize the structure of insulin receptor. (PMID: 30356040)

- line 163: “homo-complex” should be “holo-complex”.

Corrected.

- line 210: “deceased” should be “decreased”.

Corrected

- line 211: first introduction fo the activation assays with various mutants; please introduce a bit more the experiment to make it clear how it was done.

Thanks Dr. Felix for the suggestion. We have modified the original sentence to: “To further confirm the functional significance of interface I, we mutated the residues Lys47, Lys91, Phe112, and His114 in the N domain of HGF to either glutamate or alanine. HGF WT and mutants were purified and applied to H1299 cells for 10 minutes at 37 °C to test their abilities in inducing c-MET activation.”

- line 257: Extended Fig.6 doesn't exist, there are two Extended Fig.7 in the documents available for the review.

We have changed the figure number.

- line 261: “interface I” should be “interface IV”.

Corrected.

- Figure 1b: I would remove the “grey membrane” as the construct used lacks TM and kinase domains. I would replace it with a “cartoon” zipper.

Point accepted. The Figure 1b has been modified.

- Figure 4: The panels need to be rearranged; the current reading order is a, d, b and c.

Point accepted. The Figure 4 has been rearranged.

- Cryo-EM data table: there is an extra “/” between “c-MET” and “II” on the third column of data.

Corrected.

Felix Weis

Reviewer #3 (Remarks to the Author):

This manuscript by Uchikawa et al. addresses a long-standing question of how HGF and related ligands are able to activate the c-MET receptor tyrosine kinase. This is a very relevant question given the multiple oncogenic roles c-MET plays in human cancers. The authors use cryo-EM to explore the stoichiometry of ligand binding, the composition of the ligand-receptor interfaces, the role of heparin sulphate (a surrogate GAG) and differences between members of the same ligand family in binding c-MET. This study reveals how a single HGF can dimerise two c-MET receptors, while a second HGF molecule plays an auxiliary role in strengthening the receptor dimer. The close proximity of two heparan sulphate GAG binding sites on HGF-1/HGF2 and c-MET II suggests HS stabilises the c-MET dimer conformation. There is an excellent comparison between HGF and NK1 ligands, the former ligand drives an asymmetric 2:2 complex while the latter stabilises a symmetric C2 2:2: complex. The study is convincing and technically sound with clear and well-presented figures. Overall, it's a very compelling story with novelty and the authors are to be congratulated on answering a long-standing question. It is likely to drive future work to explore differences in NK versus HGF driven signalling through c-MET. I would therefore recommend this study for publication. I have gathered together some typos in the manuscript below and raise some questions and minor points that may improve the manuscript clarity.

We thank the reviewer for their positive assessment of our work.

Minor points

1. A few typos I noticed;

Line 108 – features

Corrected.

Line 131 – absent

Corrected

Line 144 – holo-complex? Also elsewhere in the manuscript, line 146 etc.

Corrected

Line 208 – interacts "with"

Corrected

Line 210 – decreased

Corrected

Line 228 – deficiency

Corrected

Line 276 – proteolysis

Corrected

Line 286 – to a certain level

Corrected

Line 332 – by the same set

Corrected

Line 385 – unstable

This part has been removed from the text based other reviewers' suggestions.

Line 395 – strengthens

Corrected

Line 405 – Such a unique

Corrected

Line 408 – heparin molecule

Corrected

Line 411 – heparan molecule could potentially

Corrected

Line 413 – heparan polymers

Corrected

Line 421 – which to a certain extent could explain

Corrected

Line 736 – Heparin strengthens

Corrected

Line 804 – may be better to say "(d) Modular structure of of HGF"

Corrected

2. Heparin sulphate (Tinzaparin sodium - an anticoagulant) is used as a surrogate for heparan sulphate GAG presented by proteoglycans. It may be good to mention this at some point in case the casual reader thinks it is heparin that participates in c-MET activation.

Good point! In the revised manuscript, from line 114: "Tinzaparin sodium, a low molecular weight heparin surrogate, was also added to enhance the binding affinity between c-MET and HGF."

3. Figure 2 and 3 improvements – it would help if the location of the HGF or c-MET mutations were emphasised in Figures 2 and 3 for clarity, either a box around the residues mutated or underlined would help. I would indicate that panel 2f is targeting HGF residues, whereas Figure 2g panel targets c-MET residues. I would indicate which interface residues belong to in panel 2g.

These are great suggestions. We have labelled the mutated HGF and c-MET residues with rectangular boxes. We have indicated that panel 2f targets HGF residues, while Figure 2g panel targets c-MET residues (close to left side of the horizontal axis).

4. In view of this study it may (or may not) be interesting to comment on a reinterpretation of the high and low affinity binding sites for HGF originally found for cells transfected with c-MET, proposed to be the high affinity receptor with heparin/heparan as the low affinity receptor.

Heparin alone is unlikely able to serve as the receptor for HGF. C-MET is the well established receptor for HGF, while heparin facilitates the formation of the c-MET/HGF complex.

5. It would be helpful to mention where the pivot point is for c-MET stalk region conformational change is located in Extended Data Figure 4? Is it within a domain or within a connecting linker region?

Good point! The pivot point is within the short linker connecting the PSI and IPT1 domains. We have mentioned this point in the revised manuscript.

6. It would be good to clarify that pEZT-BM is a Bacmam vector that can be used to infect both HEFK293F cells as well as Sf9 insect cells.

We have clarified this in the method part of revised manuscript.

7. It may be helpful to clarify in line 822, what is meant by actin1 and actin2 used for the normalisation of c-MET phosphorylation.

We run separate gels to probe c-MET and phospho-c-MET. Actin1 is the loading control for phospho-c-MET, while Actin2 is the loading control for c-MET. We have clarified this in the figure legend of Extended Data Figure 5

REVIEWERS' COMMENTS

Reviewer #1 (Remarks to the Author):

The authors haven done a great job at revising their manuscript. All of my issues (and I think also those of the other reviewers) have been adequately addressed.

Reviewer #2 (Remarks to the Author):

The authors answered to all my concerns; the manuscript reads well and the story described is extremely appealing. I therefore strongly recommend this study for publication.

Felix Weis

Reviewer #1 (Remarks to the Author):

The authors haven done a great job at revising their manuscript. All of my issues (and I think also those of the other reviewers) have been adequately addressed.

Thanks this reviewer for the positive comments.

Reviewer #2 (Remarks to the Author):

The authors answered to all my concerns; the manuscript reads well and the story described is extremely appealing. I therefore strongly recommend this study for publication.

Felix Weis

Thanks Dr. Weis for the positive comments.